# An algorithm to retrieve ice water content profiles in cirrus clouds from the synergy of ground-based lidar and thermal infrared radiometer measurements

Friederike Hemmer[1*], Laurent C.-Labonnote[1], Frédéric Parol[1], Gérard Brogniez[1], Bahaiddin Damiri[2], and Thierry Podvin[1]

[1]Laboratoire d'Optique Atmosphérique, Université de Lille, Villeneuve d'Ascq, France
[2]Cimel Electronique, Paris, France
[*]Now at: Laboratoire de Météorologie Dynamique/Institut Pierre-Simon Laplace (LMD/IPSL), Sorbonne Université, Ecole Polytechnique, CNRS, Paris, France

**Correspondence:** Friederike Hemmer (friederike.hemmer@lmd.jussieu.fr)

**Abstract.** The algorithm presented in this paper was developed to retrieve ice water content (IWC) profiles in cirrus clouds. It is based on optimal estimation theory and combines ground-based visible lidar and thermal infrared (TIR) radiometer measurements in a common retrieval framework in order to retrieve profiles of IWC together with a correction factor for the backscatter intensity of cirrus cloud particles. As a first step, we introduce a method to retrieve extinction and IWC profiles in cirrus clouds from the lidar measurements alone and demonstrate the shortcomings of this approach due to the backscatter-to-extinction ambiguity. As a second step, we show that TIR radiances constrain the backscattering of the ice crystals at the visible lidar wavelength by constraining the ice water path (IWP) and hence the IWC which is linked to the optical properties of the ice crystals via a realistic bulk ice microphysical model. The scattering phase function obtained from the microphysical model is flat around the backscatter direction (i.e. there is no backscatter peak). We show that using this flat backscattering phase function to define the backscatter-to-extinction ratio of the ice crystals in the retrievals with the lidar only algorithm results in an overestimation of the IWC which is inconsistent with the TIR radiometer measurements. Hence, a synergy algorithm was developed that combines the attenuated backscatter profiles measured by the lidar and the measurements of TIR radiances in a common optimal estimation framework to retrieve the IWC profile together with a correction factor for the phase function of the bulk ice crystals in backscattering direction. We show that this approach yields consistent lidar and TIR results. The resulting lidar ratios for cirrus clouds are found to be consistent with previous independent studies.

## 1 Introduction

The importance of clouds for the climate system has been extensively discussed during the last decades (Stephens, 2005). Although their essential role in the Earth's radiative budget is unquestionable, they still remain a major source of uncertainty in climate change estimates (Boucher et al., 2013). In particular, the important but complex impact of cirrus clouds has long been recognized (Liou, 1986) but is still not well quantified. This is due to the large range of varying shapes and sizes of the ice crystals observed in cirrus clouds which may interact in different ways with atmospheric radiation by scattering and absorption

processes. The net radiative effect of cirrus clouds is generally positive but can be negative as well (Zhang et al., 1999). It is determined on the one hand by the macrophysical cloud properties, e.g. altitude, geometrical thickness, temperature and the difference between the temperature of the cloud and the surface (e.g., Stephens and Webster, 1981). On the other hand it depends on the optical properties of the cloud which are in turn governed by the microphysics, especially the size, shape and

number density of particles.

Recent advances in satellite observational systems, particularly the Cloud Profiling Radar (CPR; Stephens et al. (2002, 2008)) aboard CloudSat and the Cloud-Aerosol Lidar with Orthogonal Polarization (CALIOP; Winker et al. (2009, 2010)) aboard CALIPSO as part of the international satellite constellation known as A-Train, have shown that the occurrence of cirrus clouds in the atmosphere is much higher than previously presumed (Stubenrauch et al., 2013). Mace et al. (2009) quantified the

global occurrence frequencies to 40-60% whereas earlier estimates expected about 20-30% with a higher coverage of 60-70% in the tropics (e.g., Liou, 1986; Wylie et al., 1994). This underlines the importance of studying the characteristics of cirrus clouds to estimate their influence on the radiation budget.

Lidar measurements have proven to be a powerful tool to study even the most tenuous cloud layers (e.g., Sassen, 1991). The measured backscatter profiles provide an information about the cloud base and top altitudes which can be related to temperature

by using atmospheric temperature profiles from model reanalysis or radiosounding. Furthermore, these measurements yield the possibility to retrieve profiles of particle extinction and hence the optical depth of the cirrus cloud by making assumptions about the so-called backscatter-to-extinction ratio. However, there are more advanced lidar systems which do not require such assumptions that can introduce large errors in the estimated cloud optical depth. Raman lidars, for example, can provide particle extinction directly since the inelastic Raman backscatter signal is only sensitive to extinction but not to particle backscattering

(Ansmann et al., 1990, 1992). High-spectral resolution lidars are also capable of measuring particle extinction directly with the help of two channels: one that measures the backscatter originating from the entire atmosphere (molecular plus particle) and one that measures only the molecular contribution by removing the central portion of the signal which is associated with aerosol or cloud particles with a filter. From these two simultaneous measurements the particle extinction can be derived from the change in the slope of the molecular signal relative to a clear sky atmosphere (Turner and Eloranta, 2008). Other lidars,

e. g. CALIOP, include polarization measurements from which the cloud phase can be determined since ice crystals tend to depolarize the incident visible radiation whereas for water droplets no such depolarization is observed (Sassen, 1991). However, most ground-based lidars operated at present are simpler systems (Campbell et al., 2015). Thus, the algorithm presented here was developed for the exploitation of data from a simple micro-pulse lidar (combined with thermal infrared (TIR) radiance measurements) although it might be applied to other lidars in future studies. Our method should be applicable to combined TIR

and simple backscatter lidar measurements from ground-based as well as space-based observations. Concerning cirrus clouds, there already exist a large number of lidar studies dealing with their occurrence frequencies and characteristics, either based on satellite data (e.g., Berthier et al., 2008; Campbell et al., 2015), ground-based data (e.g., Ansmann et al., 1993; Keckhut et al., 2006; Giannakaki et al., 2007; Seifert et al., 2007; Liu et al., 2015) or a combination of both (e.g., Pandit et al., 2015; Córdoba-Jabonero et al., 2017). Recently, Campbell et al. (2016) went even further and characterized the daytime radiative forcing

of cirrus clouds at the Top-of-the-Atmosphere from ground-based lidar measurements at a mid-latitude site and underlined thereby the importance of ground-based measurements for the estimation of the radiative effect of cirrus.

In spite of their undenied importance, lidar systems are not the only tools to study cirrus clouds. Another important source of information are measurements from passive TIR radiometers which are also performed from the ground or from space,
e.g. the Imaging Infrared Radiometer (IIR) aboard CALIPSO (Winker et al., 2003) or the MODerate resolution Imaging Spectroradiometer (MODIS) aboard Aqua/Terra (King et al., 1992, 2003). These measurements are sensitive to the optical and integrated properties of the cloud, for example the ice water path (IWP). A well-known method using radiances in the TIR wavelength region is the split window technique (Inoue, 1985, 1987; Parol et al., 1991) which allows to retrieve the cloud top temperature and the effective emissivity of semi-transparent cirrus clouds from two channels centered around 11 $\mu$m and
12 $\mu$m, respectively. The method is based on the fact that the Brightness Temperature Difference (BTD) of these channels is always more important for thin cirrus clouds than for thick clouds or under clear sky conditions. In addition, the BTD is sensitive to the radiative and microphysical properties of the cloud. Dubuisson et al. (2008) showed by conducting radiative transfer calculations with different ice crystal models that it is possible to retrieve microphysical properties of cirrus clouds from passive TIR radiometer measurements alone.

However, in recent years synergistic approaches using independent sets of measurements in a common retrieval framework have become more and more popular. Examples that could be cited here are the raDAR/liDAR (DARDAR) algorithm to retrieve ice cloud properties from the synergy of the CPR and CALIOP measurements (Delanoë and Hogan, 2008, 2010) or the multilayer algorithm to retrieve ice and liquid water cloud properties simultaneously from three TIR radiances measured by the IIR and two MODIS reflectances measured at 0.85 $\mu$m and 2.13 $\mu$m (Sourdeval et al., 2015, 2016). There are also a
few methods combining lidar and TIR radiometer measurements that have been developed in the past. The Lidar and Infrared Radiometric (LIRAD) method introduced by Platt (1973, 1979) was the first method that combined lidar and infrared radiometer data to retrieve optical properties of cirrus clouds. It has been applied and further developed in several following studies (e.g., Platt et al., 1987; Comstock and Sassen, 2001; Platt et al., 2002). In this approach, the lidar backscatter coefficient is related theoretically to the infrared volume absorption coefficient. The emissivity of the cloud is then derived in an iterative
process by calculating a theoretical cloud radiance which is compared to the infrared radiometer measurement and adjusting the backscatter-to-extinction ratio until the theoretical and measured radiances converge. Other studies focused on the combination of lidar measurements and the split-window technique to improve the retrievals of cloud properties from passive sensors alone by integrating the information provided by the active lidar measurements in the radiative transfer calculations. Chiriaco et al. (2004) showed the theoretical potential of this approach to improve the particle size retrieval from the instruments aboard
CALIPSO, i.e. the IIR and CALIOP, and Garnier et al. (2012, 2013) developed an algorithm based on this idea to retrieve the effective emissivity, optical depth, effective diameter and IWP from CALIPSO measurements. Nevertheless, in their approach the retrieval is based on the split-window technique where information from the lidar such as scene identification and cloud altitude have been integrated. Saito et al. (2017) recently demonstrated a method to infer simultaneously the IWP, the cloud effective radius, the surface temperature and two morphological parameters, namely the fraction of plates and the surface
roughness of ice crystal aggregates, from a synergistic approach based on optimal estimation. They used the layer-integrated

total attenuated backscatter and the depolarization ratio at 532 nm from CALIOP as well as the brightness temperatures at 8.65, 10.6 and 12.0 μm from the IIR in a common retrieval framework to obtain the parameters cited above.

The algorithm proposed in this paper also establishes a synergy between lidar and TIR radiometer measurements although our lidar is a simple micro-pulse lidar and does not possess depolarization channels. In contrast to Saito et al. (2017), we use the whole backscattering profile measured by the lidar together with two TIR radiances in the measurement vector to retrieve profiles of particle extinction and ice water content (IWC). This allows us to include the profile information from the active lidar measurements in the radiative transfer calculations in the TIR which is an improvement since common retrieval algorithms often assume plane-parallel and homogeneous conditions. As a first step, we developed an algorithm to retrieve extinction and IWC profiles in thin cirrus clouds from ground-based lidar measurements alone. This algorithm is based on the method of Stephens et al. (2001) who used an optimal estimation approach to invert the lidar equation to retrieve profiles of particle extinction from space-borne data collected during the Lidar in Space Technology Experiment (LITE, McCormick et al. (1993)). To overcome the backscatter-to-extinction ambiguity arising from the combination of scattering and absorption processes when regarding the lidar measurements alone, Stephens et al. (2001) introduced an optical depth constraint in form of an additional measurement. In contrast to this approach, we developed in a second step a synergy algorithm that integrates actual measurements of TIR radiances in the optimal estimation framework. We will show that these radiances constrain the backscattering of the ice crystals at the visible lidar wavelength by constraining the IWP and hence the IWC which is linked to the optical properties of the ice crystals via the microphysical ice cloud model of Baran et al. (2001, 2014a, b) and Vidot et al. (2015).

The paper is organized as follows: Section 2 briefly introduces the instruments and data used in this study. Section 3 presents our approach to the lidar retrieval problem and describes the algorithm for the retrieval of extinction and IWC profiles from lidar measurements as well as the underlying microphysical model for cirrus clouds. In this section, we also discuss the above-mentioned backscatter-to-extinction ambiguity before the ability of TIR radiances to constrain the backscatter-to-extinction ratio is outlined. Section 4 presents the new algorithm using the synergy of lidar and TIR radiances which has been developed on the basis of the lidar only algorithm. Finally, Sect. 5 concludes this study.

## 2   Instrumentation and data

The data used in this study is originating from the measurement platform of the Laboratoire d'Optique Atmosphérique (LOA) situated on the campus of the University of Lille in northern France. This platform is equipped with, amongst other instruments, an elastic-backscatter micro-pulse lidar and a TIR radiometer.

The lidar is a Cloud and Aerosol Microlidar type CAML-CE370 (Pelon et al., 2008) developed by the company CIMEL Electronique. It is an eye-safe lidar system which operates at a single wavelength of 532 nm and does not include depolarization. The system is automated and has been operated continuously since 2007, hence a large archive of data is available for the LOA measurement site. The type of laser integrated in the instrument is a frequency doubled Nd:YAG laser. The divergence of the laser beam as well as the field of view (FOV) of the receiver are both 55 μrad. The pulse duration is 100 ns and the repetition

rate 4.7 kHz which results in a vertical resolution of 15 m defined by $100\,\text{ns} \cdot c/2$ where $c$ is the speed of light. The lidar profiles used in this study are averaged over one minute and a vertical binomial filter has been applied in order to smooth the signal. The lidar was pointing directly vertical with a zenith angle of $0°$.

The radiometer is called Conveyable Low-Noise Infrared Radiometer for Measurements of Atmosphere and Ground Surface Targets (CLIMAT) (Sicard et al., 1999; Legrand et al., 2000; Brogniez et al., 2003). It was developed to measure radiances in the TIR wavelength region in three different spectral bands centered at $8.7\,\mu\text{m}$, $10.8\,\mu\text{m}$ and $12.0\,\mu\text{m}$. The full width at half maximum (FWHM) of each of these channels is $1\,\mu\text{m}$. In the following, we will call them C09, C11 and C12, respectively. There are two different versions of the instrument: one was designed for ground-based measurements and the other one for airborne measurements. Although we exploit ground-based observations in this study, the CLIMAT instrument currently installed on the LOA measurement platform is of aircraft type. This instrument measures the radiances of the three different channels simultaneously and has a FOV of $3.5°$. It consists of two main parts: the optical head containing the optical elements as well as the detector, and the control unit containing the electronics and the memory. The main optic consists of two germanium lenses: the objective which is a standard convex-plane lens, and the condenser which is a "best-shaped" meniscus designed to minimize the geometrical aberrations (Legrand et al., 2000). The condenser is situated in the focal plane of the objective. The optical head is constructed respecting the so-called Köhler design which means that the detector is located in the conjugate plane of the objective with respect to the condenser. The radiation is measured by a thermopile where the hot junction is heated by the incident radiation and the temperature of the cold junction is determined by the ambient temperature of the cavity. In contrast to the lidar system, the CLIMAT instrument is operated manually depending on the weather conditions.

It should be noted that due to the larger FOV of the TIR radiometer compared to the FOV of the lidar, the two instruments do not see exactly the same cloud area. This difference also depends on the altitude of the cloud. As in almost all remote sensing algorithms, we assumed a homogeneous cloud in the instrument FOV and did not take into account any uncertainty due to sub-pixel heterogeneity.

## 3   Lidar Only algorithm

The algorithm presented here was developed to retrieve profiles of particle extinction and IWC from measured lidar backscattering profiles. We propose a method to retrieve simultaneously a profile of aerosol extinction in the layers close to the ground and a profile of IWC inside cirrus layers. However, our main focus is the characterization of the microphysical properties of cirrus clouds. There are many techniques to invert the lidar equation including the classical Klett-Fernald method (Klett, 1981, 1985; Fernald, 1984). Our algorithm closely follows the method described by Stephens et al. (2001) which is based on optimal estimation theory introduced by Rodgers (1976, 1990, 2000) which is now a common approach for the inversion of remote sensing data. One advantage of this approach is that it directly provides an estimation of the uncertainties together with the retrieved quantities. Furthermore, it facilitates the introduction of additional information in a common retrieval framework in order to constrain the retrieved parameters. The additional information could be, for example, measurements of polarization or

at other lidar wavelengths, as well as TIR radiometer measurements as will be discussed in Sect. 4 of this paper. As a first step, we focus in this section on the retrieval of extinction and IWC profiles from the lidar measurements alone.

## 3.1 Lidar retrieval problem and microphysical assumptions

The relationship between the range-resolved backscattered power, $P(r)$, and the atmospheric scattering and attenuation properties is described by the lidar equation and may be expressed as follows:

$$C \cdot P(r)r^2 = (\beta_m(r) + \beta_p(r)) \cdot \exp\left[-2 \int\limits_0^r (\sigma_m(r') + \eta\sigma_p(r'))\mathrm{d}r'\right],\tag{1}$$

where $C$ is a calibration constant depending on the lidar system and the atmospheric profile. $\beta(r)$ and $\sigma(r)$ represent the backscattering and extinction coefficients, respectively, and both contain a contribution arising from purely molecular backscattering ($\beta_m(r)$) or extinction ($\sigma_m(r)$) and a contribution arising from cloud or aerosol particles that may be present in the atmosphere ($\beta_p(r)$ and $\sigma_p(r)$, respectively). The factor $\eta$ accounts for multiple scattering processes. For the remainder of this paper we assume that $\eta = 1$ for aerosols and $\eta = 0.75$ for cirrus clouds. The value of 0.75 for cirrus clouds has been chosen based on the PhD thesis of Nohra (2016) where the multiple scattering factor has been evaluated by comparing the optical depth retrieved from measurements of the micro-pulse lidar in Lille to the optical depth retrieved from CALIOP and adjusting $\eta$ to find a coherent retrieval. The multiple scattering factor used for the CALIOP version 3 retrievals is $\eta = 0.6$ (Garnier et al., 2015). For ground-based lidars the multiple scattering effect is less important since they have a much smaller FOV in combination with a shorter distance to the cloud, although it should not be neglected because large ice crystals may considerably increase the forward scattering of the laser beam (Donovan and van Lammeren, 2001). However, since our knowledge of this parameter is rather poor we assign a large error to it in our optimal estimation algorithm (see Sect. 3.2).

The retrieval of particle optical properties from elastic lidar measurements alone is challenging since there is an intrinsic ambiguity between the effects of backscattering and extinction arising from the combination of scattering and absorption processes in the atmosphere. In Eq. 1, the backscattering coefficient of aerosol or cloud particles, $\beta_p(r)$, can be replaced by

$$\beta_p(r) = k(r) \cdot \sigma_p(r),\tag{2}$$

where $k(r)$ represents the range-dependent backscatter-to-extinction ratio. Since we are using a simple micro-pulse lidar, we need to introduce some assumptions for $k(r)$ to retrieve profiles of extinction by aerosol and cloud particles. Unfortunately, this parameter is highly variable and depends strongly on the type, size and shape of the atmospheric particles.

In this study we are focusing on the retrieval of cirrus cloud properties. Thus, the backscatter-to-extinction coefficient for aerosols is assumed to be constant and is fixed to 64 sr. This value originates from the Optical Properties of Aerosols and Clouds (OPAC) database (Hess et al., 1998b) and corresponds to a water soluble urban aerosol. Since our measurement site is located in an urban/industrial area, we used the optical properties for this aerosol type (for the visible lidar wavelength as well as for the TIR as will be discussed in Sect. 4.1) from the OPAC database to test our new algorithm. This parameter needs to be refined in future studies depending on the aerosol type which is actually present during the measurement obtained from

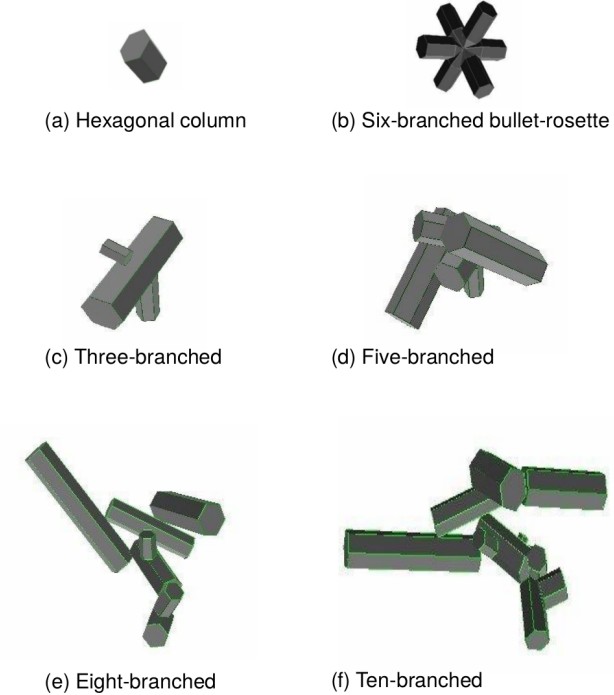

**Figure 1.** The ensemble model of Baran and Labonnote (2007). (a) Hexagonal ice column which represents the smallest member, (b) Six-branched bullet-rosette, (c) Three-branched ice crystal, (d) Five-branched ice crystal, (e) Eight-branched ice crystal, and (f) Ten-branched ice crystal which represents the largest member (courtesy of A. Baran).

additional information. However, as discussed above for the multiple scattering factor, we also assign a large uncertainty to the lidar ratio of aerosols to account for our rather poor knowledge of it. Unfortunately, this reduces the quality of the information returned by our algorithm.

The backscatter-to-extinction ratio for cirrus clouds is calculated using the definition of Mishchenko et al. (1997):

$$k = \varpi_0 \cdot P_{11}(\pi), \tag{3}$$

where $\varpi_0$ is the particle single scattering albedo and $P_{11}(\pi)$ the phase function in the exact backscattering direction.

We obtain the single scattering properties (scattering coefficient, absorption coefficient and asymmetry parameter) for each cloud layer from the parametrization of Vidot et al. (2015) which is based on the ensemble model for cirrus introduced by Baran and Labonnote (2007). The idea of this model is to represent the variability of ice crystal sizes and shapes inside a cirrus cloud by assuming a distribution of some idealized shapes rather than assuming just a single geometrical form throughout the whole size spectrum. Observed ice crystal shapes in cirrus clouds range from simple pristine particles such as hexagonal ice columns and bullet-rosettes (associated with small particles) over aggregates of these particles to aggregate chains while the complexity of the crystal tends to increase with increasing size. The ensemble model of Baran and Labonnote (2007) attempts

to reproduce these observations by using the six members shown in Fig. 1. The smallest ice crystals are represented by the first two members, which are simple hexagonal ice columns and bullet-rosettes. The following members represent larger and more complex ice crystals by arbitrarily attaching up to ten hexagonal elements to create chain-like structures. For the calculation of the bulk optical properties, the particle size distribution (PSD) of Field et al. (2005, 2007) is assumed which is independent of assumptions about the ice crystal shape and depends only on the in-cloud temperature and the IWC. This parametrization has been constructed based on a large number of in situ measured PSDs and does not include measurements of ice crystal sizes less than $100\,\mu m$ due to the shattering problem (Strapp et al., 2001; Field et al., 2003). For particles smaller than $100\,\mu m$ an exponential PSD is assumed. Baran et al. (2011, 2014a, b) used the Field et al. (2005, 2007) parametrization to calculate the single scattering properties for the ensemble model as functions of IWC and in-cloud temperature for a total number of 20662 in situ measurements from different aircraft-based field campaigns located in the tropics and in the mid-latitudes. They created a database of optical ice cloud properties comprising 145 wavelengths between 0.2 and $120\,\mu m$. This database was used by Vidot et al. (2015) to develop a new ice cloud parametrization that predicts the single scattering properties named above as functions of in-cloud temperature and IWC without the need of a priori information on the shape and the effective diameter of the ice crystals. For the remainder of this paper, we will call this microphysical model BV2015.

Since our algorithm seeks to retrieve the IWC for cirrus cloud layers, the extinction required in the lidar equation (Eq. 1) is calculated from the scattering and absorption coefficients obtained from the BV2015 parametrization as a function of the IWC of each cloud layer. The necessary temperature information is obtained from the European Centre for Medium-Range Weather Forecasts (ECMWF) reanalysis by matching atmospheric temperature profiles to the corresponding cirrus cloud altitude. The single scattering albedo required in Eq. 3 is also calculated from the scattering and absorption coefficients, and the scattering phase function is generated from the asymmetry parameter using the analytic phase function of Baran et al. (2001) which is a linear piecewise parametrization of the Henyey-Greenstein phase function depending only on the asymmetry parameter. It is kept smooth and featureless since atmospheric ice crystals may be distorted, roughened or contain inclusions of air bubbles or aerosols. All these processes would remove or reduce the optical features of the phase function like the halos at $22°$ or $46°$ and the backscattering peak (e.g., Macke et al., 1996; C.-Labonnote et al., 2001). It has been demonstrated by the authors that the Baran et al. (2001) parametrization reproduces short-wave multi-angle satellite and aicraft observations, and Baran and Francis (2004) showed a good agreement with high-resolution infrared observations between 3 and $18\,\mu m$. It has also been shown to be in good agreement with the backscattering features observed from POLarization and Directionality of the Earth's Reflectances (POLDER) measurements (Baran and Labonnote, 2007). The scattering phase function, especially in the exact backscattering direction, is a crucial parameter in our algorithm since it defines the lidar backscatter-to-extinction ratio. It will be discussed in more detail in Sect. 3.3.1.

## 3.2 Inversion method

As mentioned above, we apply an optimal estimation method to invert the lidar equation following Stephens et al. (2001). Optimal estimation is based on a Bayesian approach which uses probability density functions to link the measurement space to the state space accounting for their uncertainties (Rodgers, 2000). This approach allows finding the most likely solution which

is consistent with both the measurement and any given prior knowledge of the state within the range of their uncertainties. In general, the measurement vector $\boldsymbol{y}$ can be related to the state vector $\boldsymbol{x}$ via the forward model $\mathbf{F}$ by

$$\boldsymbol{y} = \mathbf{F}(\boldsymbol{x}) + \boldsymbol{\epsilon}, \tag{4}$$

where $\boldsymbol{\epsilon}$ represents the uncertainties arising from the measurements and the forward model. The aim of every inversion method

is to invert the connection between the state vector and the measurement vector which is given by the forward model in order to retrieve the elements of the state vector using the information provided by the measurement vector.

Following Rodgers (2000), the best estimation of the state vector can be obtained by minimizing the following cost function

$$\Phi = [\boldsymbol{y} - \mathbf{F}(\boldsymbol{x})]^T \mathbf{S}_\epsilon^{-1} [\boldsymbol{y} - \mathbf{F}(\boldsymbol{x})] + [\boldsymbol{x} - \boldsymbol{x}_a]^T \mathbf{S}_a^{-1} [\boldsymbol{x} - \boldsymbol{x}_a]. \tag{5}$$

There are two contributions in this cost function: the first term on the right-hand side of Eq. 5 represents the contribution arising from the forward model and the measurement where $\mathbf{S}_\epsilon$ is the sum of the variance-covariance matrices of the forward model and the measurement, and the second term represents the contribution from the so-called a priori which contains the prior knowledge of the state vector before the measurement has been performed. $\mathbf{S}_a$ is the variance-covariance matrix of the a priori which, in our case, was chosen sufficiently large to reduce the influence of the a priori on the final retrieval.

To find the best estimate of the state vector $\hat{\boldsymbol{x}}$ that minimizes the cost function $\Phi$, an iterative method was applied following the approach of Levenberg-Marquardt (Levenberg, 1944; Marquardt, 1963) which is described in detail by Rodgers (2000). This approach is based on the Newton-Gauss method to which the parameter $\gamma$ is added that regulates the size of each iteration step in order to diminish the cost function compared to the previous step. The equation for this iteration may be expressed by

$$\boldsymbol{x}_{i+1} = \boldsymbol{x}_i + [(1+\gamma)\mathbf{S}_a^{-1} + \mathbf{K}_i^T \mathbf{S}_\epsilon^{-1} \mathbf{K}_i]^{-1} \{\mathbf{K}_i^T \mathbf{S}_\epsilon^{-1} [\boldsymbol{y} - \mathbf{F}(\boldsymbol{x}_i)] - \mathbf{S}_a^{-1}[\boldsymbol{x}_i - \boldsymbol{x}_a]\}, \tag{6}$$

where $\mathbf{K}$ is the Jacobian containing the sensitivities of each of the parameters of the state vector to each individual measurement. $\mathbf{K}$ acts as a matrix of weights in Eq. 6 and may also be referred to as Weighting Matrix or Kernel. When convergence is reached, the variance-covariance matrix of the retrieved state vector $\hat{\boldsymbol{x}}$ is given by

$$\mathbf{S}_{\hat{x}} = (\mathbf{S}_a^{-1} + \mathbf{K}^T \mathbf{S}_\epsilon^{-1} \mathbf{K})^{-1}, \tag{7}$$

where $\mathbf{K}$ and $\mathbf{S}_\epsilon$ correspond to the last iteration. The matrix $\mathbf{S}_{\hat{x}}$ allows to identify the error on each retrieved parameter.

Convergence is obtained when the following convergence test is true

$$[\boldsymbol{y} - \mathbf{F}(\hat{\boldsymbol{x}})]^T \mathbf{S}_\epsilon^{-1} [\boldsymbol{y} - \mathbf{F}(\hat{\boldsymbol{x}})] < \mathrm{N}, \tag{8}$$

where N is the number of elements in the measurement vector.

The application of this theoretical framework to the lidar retrieval problem requires the definition of all necessary elements described above. The state vector $\boldsymbol{x}$ contains the desired quantities to be retrieved. These are in our case a profile of extinction

(denoted by $\sigma_p$) outside the cirrus cloud and a profile of IWC inside the cloud layer,

$$\boldsymbol{x} = [\sigma_p(r_1), \sigma_p(r_2), \ldots, \sigma_p(r_{j\_\mathrm{bot}-1}), \mathrm{IWC}(r_{j\_\mathrm{bot}}), \ldots, \mathrm{IWC}(r_{j\_\mathrm{top}}), \sigma_p(r_{j\_\mathrm{top}+1}), \ldots, \sigma_p(r_N)]^T, \tag{9}$$

where the subscripts $j\_bot$ and $j\_top$ denote the range index of the bottom cloud layer and the top cloud layer, respectively. The measurement vector $\boldsymbol{y}$ consists of the logarithm of the calibrated range-corrected lidar signal,

$$\boldsymbol{y} = \left[\ln(C \cdot P(r_1)r_1^2), \ln(C \cdot P(r_2)r_2^2), \dots, \ln(C \cdot P(r_N)r_N^2)\right]^T. \tag{10}$$

As mentioned in Sect. 2, the measurements provide an information about the backscattering particles every 15 m and the same constant vertical resolution is used in the state vector.

Following Stephens et al. (2001), the forward model $\mathbf{F}$ is given by the lidar equation in its logarithmic and discretized form:

$$\mathrm{F}(x_j, b_j) = \ln(\beta_m(r_j) + k(r_j)\sigma_p(r_j)) - 2\sum_{l=1}^{j}\left[\bar{\sigma}_{m,l} + \eta\bar{\sigma}_{p,l}\right]\Delta R, \tag{11}$$

defined at each range $r_j$ where $j = 2, ..., N$. The overline indicates layer mean values. As discussed in Sect. 3.1, the multiple scattering factor $\eta$ is set to unity for aerosols and 0.75 for cirrus clouds, and $\Delta R$ is the range resolution of 15 m of the lidar system. The state vector $\boldsymbol{x}$ defined in Eq. 9 contains the IWC inside the cloud, hence the extinction $\sigma_p(r_j)$ which is required in Eq. 11 is calculated as a function of IWC, $\sigma_p(\mathrm{IWC}(r_j))$, for all $r_j$ inside the cloud layer with the BV2015 microphysical model described in Sect. 3.1. Vector $\boldsymbol{b}$ in Eq. 11 represents the non-retrieved parameters and is defined below (see Eqs. 16 to 18).

The Jacobian $\mathbf{K}$ contains the sensitivities of the forward model to each element of the state vector,

$$\mathbf{K} = \begin{pmatrix} \frac{\partial \mathrm{F}_1}{\partial \sigma_{p,1}} & \cdots & \frac{\partial \mathrm{F}_1}{\partial \sigma_{p,j\_bot-1}} & \frac{\partial \mathrm{F}_1}{\partial \mathrm{IWC}_{j\_bot}} & \cdots & \frac{\partial \mathrm{F}_1}{\partial \mathrm{IWC}_{j\_top}} & \frac{\partial \mathrm{F}_1}{\partial \sigma_{p,j\_top+1}} & \cdots & \frac{\partial \mathrm{F}_1}{\partial \sigma_{p,N}} \\ \frac{\partial \mathrm{F}_2}{\partial \sigma_{p,1}} & \cdots & \frac{\partial \mathrm{F}_2}{\partial \sigma_{p,j\_bot-1}} & \frac{\partial \mathrm{F}_2}{\partial \mathrm{IWC}_{j\_bot}} & \cdots & \frac{\partial \mathrm{F}_2}{\partial \mathrm{IWC}_{j\_top}} & \frac{\partial \mathrm{F}_2}{\partial \sigma_{p,j\_top+1}} & \cdots & \frac{\partial \mathrm{F}_2}{\partial \sigma_{p,N}} \\ \vdots & \cdots & \vdots & \vdots & \cdots & \vdots & \vdots & \cdots & \vdots \\ \frac{\partial \mathrm{F}_{j\_bot-1}}{\partial \sigma_{p,1}} & \cdots & \frac{\partial \mathrm{F}_{j\_bot-1}}{\partial \sigma_{p,j\_bot-1}} & \frac{\partial \mathrm{F}_{j\_bot-1}}{\partial \mathrm{IWC}_{j\_bot}} & \cdots & \frac{\partial \mathrm{F}_{j\_bot-1}}{\partial \mathrm{IWC}_{j\_top}} & \frac{\partial \mathrm{F}_{j\_bot-1}}{\partial \sigma_{p,j\_top+1}} & \cdots & \frac{\partial \mathrm{F}_{j\_bot-1}}{\partial \sigma_{p,N}} \\ \frac{\partial \mathrm{F}_{j\_bot}}{\partial \sigma_{p,1}} & \cdots & \frac{\partial \mathrm{F}_{j\_bot}}{\partial \sigma_{p,j\_bot-1}} & \frac{\partial \mathrm{F}_{j\_bot}}{\partial \mathrm{IWC}_{j\_bot}} & \cdots & \frac{\partial \mathrm{F}_{j\_bot}}{\partial \mathrm{IWC}_{j\_top}} & \frac{\partial \mathrm{F}_{j\_bot}}{\partial \sigma_{p,j\_top+1}} & \cdots & \frac{\partial \mathrm{F}_{j\_bot}}{\partial \sigma_{p,N}} \\ \vdots & \cdots & \vdots & \vdots & \cdots & \vdots & \vdots & \cdots & \vdots \\ \frac{\partial \mathrm{F}_{j\_top}}{\partial \sigma_{p,1}} & \cdots & \frac{\partial \mathrm{F}_{j\_top}}{\partial \sigma_{p,j\_bot-1}} & \frac{\partial \mathrm{F}_{j\_top}}{\partial \mathrm{IWC}_{j\_bot}} & \cdots & \frac{\partial \mathrm{F}_{j\_top}}{\partial \mathrm{IWC}_{j\_top}} & \frac{\partial \mathrm{F}_{j\_top}}{\partial \sigma_{p,j\_top+1}} & \cdots & \frac{\partial \mathrm{F}_{j\_top}}{\partial \sigma_{p,N}} \\ \frac{\partial \mathrm{F}_{j\_top+1}}{\partial \sigma_{p,1}} & \cdots & \frac{\partial \mathrm{F}_{j\_top+1}}{\partial \sigma_{p,j\_bot-1}} & \frac{\partial \mathrm{F}_{j\_top+1}}{\partial \mathrm{IWC}_{j\_bot}} & \cdots & \frac{\partial \mathrm{F}_{j\_top+1}}{\partial \mathrm{IWC}_{j\_top}} & \frac{\partial \mathrm{F}_{j\_top+1}}{\partial \sigma_{p,j\_top+1}} & \cdots & \frac{\partial \mathrm{F}_{j\_top+1}}{\partial \sigma_{p,N}} \\ \vdots & \cdots & \vdots & \vdots & \cdots & \vdots & \vdots & \cdots & \vdots \\ \frac{\partial \mathrm{F}_N}{\partial \sigma_{p,1}} & \cdots & \frac{\partial \mathrm{F}_N}{\partial \sigma_{p,j\_bot-1}} & \frac{\partial \mathrm{F}_N}{\partial \mathrm{IWC}_{j\_bot}} & \cdots & \frac{\partial \mathrm{F}_N}{\partial \mathrm{IWC}_{j\_top}} & \frac{\partial \mathrm{F}_N}{\partial \sigma_{p,j\_top+1}} & \cdots & \frac{\partial \mathrm{F}_N}{\partial \sigma_{p,N}} \end{pmatrix}, \tag{12}$$

where the terms $\mathrm{F}(x_j, b_j)$, $\sigma_p(r_j)$ and $\mathrm{IWC}(r_j)$ have been shortened to $\mathrm{F}_j$, $\sigma_{p,j}$ and $\mathrm{IWC}_j$, respectively, to increase the readability. This short notation will be used for all variables which are a function of range for the remainder of this article. Inside cirrus cloud layers, the partial derivatives are expressed by

$$\mathrm{K}_{ij} = \frac{\partial \mathrm{F}_i}{\partial \mathrm{IWC}_j} = \frac{\partial \mathrm{F}_i}{\partial \sigma_{p,j}}\frac{\partial \sigma_{p,j}}{\partial \mathrm{IWC}_j}, \tag{13}$$

and can be obtained from the BV2015 parametrization. The partial derivatives with respect to extinction are calculated by differentiating Eq. 11 with respect to $\sigma_p$:

$$
\mathrm{K}_{ij} = \begin{cases} 0 & \text{for } i < j \\ -2\eta\Delta R & \text{for } i > j \\ \frac{k_i}{\beta_{m,i}+k_i\sigma_{p,i}} - 2\eta\Delta R & \text{for } i = j \end{cases} \tag{14}
$$

It should be noted that in the case of opaque cirrus clouds which completely attenuate the lidar signal, the size of the measurement vector and consequently the size of the state vector are reduced. In this case, only the altitudes until full attenuation of the lidar signal are considered. Thus, the size of the Jacobian is reduced as well and it contains only $N_\mathrm{att}$ lines (and columns) where $N_\mathrm{att}$ is the number of levels until the altitude of full attenuation.

Following Stephens et al. (2001), all variance-covariance matrices are assumed to be diagonal. Hence, the variance-covariance matrix of the a priori can be defined by

$$
\mathrm{S}_{a,ii} = \sigma_{a,i}^2, \tag{15}
$$

where $\sigma_{a,i}$ are the variances of each of the elements of the a priori state vector. In this study, we have assigned sufficiently large variances to the a priori in order to mainly rely on the information contained in the measurement vector.

As mentioned above, vector $\boldsymbol{b}$ in Eq. 11 represents the non-retrieved parameters which are each a function of altitude,

$$
b_1 = \{\,\beta_{m,j}\,; j = 1,\ldots N\,\}, \tag{16}
$$

$$
b_2 = \{\,k_j\,; j = 1,\ldots N\,\}, \tag{17}
$$

$$
b_3 = \{\,\eta_j\,; j = 1,\ldots N\,\}. \tag{18}
$$

The molecular extinction $\sigma_m$ does not need to be considered as a non-retrieved parameter because it can be obtained from $\beta_m$ by multiplication with the constant molecular backscatter-to-extinction ratio of $3/8\pi$ (Fernald, 1984). The variance-covariance matrix of the forward model and the measurement is then defined by

$$
S_{\epsilon,ii} = \sigma_{y,i}^2 + \sigma_{b1,i}^2 + \sigma_{b2,i}^2 + \sigma_{b3,i}^2, \tag{19}
$$

where $\sigma_{y,i}$ represents the measurement error and $\sigma_{b1,i}$, $\sigma_{b2,i}$ and $\sigma_{b3,i}$ represent the errors on the non-retrieved parameters in the forward model calculated via

$$
\sigma_{b1,i} = \frac{p_\beta(\%)\cdot\beta_{m,i}}{\beta_{m,i}+k_i\sigma_{p,i}}, \tag{20}
$$

$$
\sigma_{b2,i} = \frac{p_k(\%)\cdot k_i\sigma_{p,i}}{\beta_{m,i}+k_i\sigma_{p,i}}, \tag{21}
$$

$$
\sigma_{b3,i} = p_\eta(\%)\cdot(-2\eta\Delta R), \tag{22}
$$

where $p_\beta(\%)$, $p_k(\%)$ and $p_\eta(\%)$ represent the percentage errors assumed for the molecular backscattering profile, the backscatter-to-extinction ratio and the multiple scattering factor, respectively. As discussed in Sect. 3.1, we chose large errors on the multiple scattering factor for ice clouds and the backscatter-to-extinction ratio for aerosols and quantified them to $p_\eta(\%) = 25\%$ and $p_{k,\mathrm{aer}}(\%) = 25\%$, respectively. The same error has been attributed to the backscatter-to-extinction ratio for ice clouds since the knowledge of the phase function in the exact backscattering direction which is used in Eq. 3 is rather poor as well ($p_{k,\mathrm{ice}}(\%) = 25\%$). The error on the molecular backscattering profile, which is obtained from the empiric equation of Flamant et al. (2008) using the atmospheric temperature and pressure profiles from ECMWF reanalysis, is set to $p_\beta(\%) = 2\%$. The error on the lidar measurement depends on the altitude since the measurement noise increases with increasing altitude. It is calculated as the standard deviation around the mean over a vertically sliding window of 20 gates.

It should be noted that the characterization of the errors related to the BV2015 parametrization is very challenging. Thus, no error for the microphysical model is currently taken into account. However, this issue needs to be addressed in future studies and an evaluation of the uncertainty arising from the Vidot et al. (2015) parametrization, which has to be integrated in future versions of our algorithm, is planned. This evaluation could be performed by comparing the single scattering properties calculated directly from the ensemble model of Baran and Labonnote (2007) with the results from the Vidot et al. (2015) parametrization. Unfortunately, this is very costly to realize for all couples of IWC and temperature, particularly since the parametrization has not been developed by us. Furthermore, the uncertainty arising from this parametrization is assumed to be smaller than 5% (A. J. Baran, personal communication). Consequently, the present study neglects an error related to the microphysical model.

### 3.3 Results and discussion

#### 3.3.1 Influence of the backscatter-to-extinction ratio on the retrieved IWC

To start the iteration, a first guess is required and in this study we chose to use the a priori as first guess. In order to reach faster convergence of the algorithm, the a priori for the layers close to the ground where aerosols are present is calculated from a one step solution of the lidar equation following the approach of Stephens et al. (2001),

$$\sigma_{p,a,i} = \left\{ \exp\left[ y_i + 2\sum_{l=1}^{i-1}(\sigma_{m,l} + \eta\sigma_{p,a,l})\Delta R \right] - \beta_{m,i} \right\}/k_i. \tag{23}$$

For the layers above the boundary layer, the algorithm converges fast enough when the molecular signal is used as a priori. For ice cloud layers, we start the iteration from a small IWC of $0.001\,\mathrm{g\,m^{-3}}$.

Figure 2 shows an example of a measured lidar profile (represented by the red lines) containing a cirrus cloud in altitudes between 8865 and 10200 m measured on November 30, 2016 at 18.18 UTC. The black lines show the calculated forward model, in Fig. 2a for the a priori and in Fig. 2b after the last iteration step. Figure 2c shows the relative difference (in %) between the forward model and the measurement after the last iteration. Since the a priori for the lowest layers has been pre-calculated based on the lidar equation, the forward model of the a priori is already close to the measurement for the layers close to the ground. After the last iteration the forward model and the measured lidar signal overlay each other almost perfectly and the

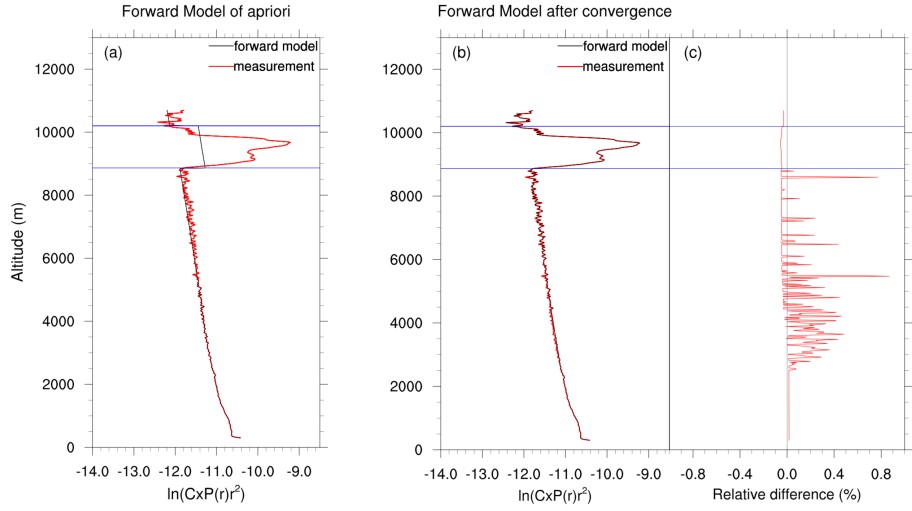

**Figure 2.** Forward model of the (a) a priori, and (b) after the last iteration step (represented by the black lines) for the lidar profile measured on November 30, 2016 at 18.18 UTC. The red lines represent the measurement, the horizontal blue lines indicate the defined cloud base and top altitudes. (c) Relative difference between the forward model and the measurement after convergence.

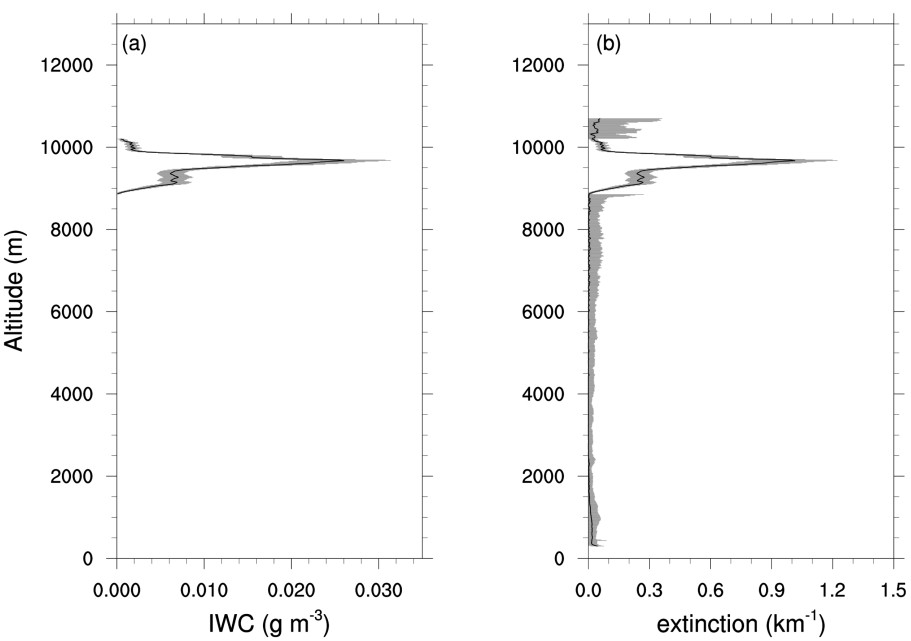

**Figure 3.** (a) Retrieved IWC profile, and (b) retrieved extinction profile for the lidar profile measured on November 30, 2016 at 18.18 UTC. Shaded areas represent the total error on the retrieved quantities.

relative difference between the forward model and the measurement is less than 1% for all altitudes. This indicates that the retrieval was successful and that the cost function has been reduced by reducing the difference between the measurement and the forward model.

Figure 3 presents the corresponding retrieved IWC (Fig. 3a) and extinction (Fig. 3b) profiles. As explained in Sect. 3.2, we retrieve the IWC for the layers containing a cirrus cloud and the particle extinction for the rest of the profile. Hence, the extinction profile inside the cloud in Fig. 3b is not retrieved directly but recalculated from the IWC using the BV2015 parametrization. In the layers close to the ground, an enhanced extinction due to aerosols can be observed. Above these layers in the middle portion of the profile, the particle extinction is close to zero because there were no or very few particles present in this zone. In the ice cloud an important increase of extinction due to the ice crystals can be observed.

The retrieval results for the whole afternoon of November 30, 2016, are shown in Fig. 4. Figure 4a shows the measured lidar signal. The cloud base and top altitudes are defined based on the threshold method described by Platt et al. (1994). Cloud-base is defined as altitude where the lidar signal increases above the clear background level and this increase is larger than $n$ times the standard deviation of the background fluctuations. As a second condition it is required that the signal continues to increase for $m$ following altitude gates to assure that sudden maxima in the signal due to measurement noise are not misinterpreted as clouds. We chose values of 4 and 5 for $n$ and $m$, respectively, which are suitable for our lidar system. The cloud-top is found in the same way but by starting the search from the far end of the measurement range (about 15 km) and moving downwards.

Figure 4b shows the retrieved extinction profiles and Fig. 4c the retrieved IWC in cirrus cloud layers. The height limit up to which the retrieval is performed depends on the profile. Since the signal above the cloud becomes noisy due to attenuation by the cloud particles, the height limit for the retrieval is defined for each profile as retrieved cloud top altitude plus 500 m. If the lidar signal is completely attenuated or if the measured power becomes negative due to noise in lower altitudes, the upper limit of the measurement vector (and hence the state vector) is fixed at the uppermost measurable layer. In Fig. 4d the Cloud Optical Thickness (COT) calculated from the retrieved extinction profile is compared to the COT derived from the transmission method introduced by Young (1995) and Chen et al. (2002), that means from the shift of the signal below and above the cloud due to the extinction of the cloud. There are phases when the COT from both methods coincides quite well, e.g. between 16.6 and 18.2 UTC. However, during this period the retrieved COT is very small. When the cloud becomes geometrically and optically thicker, our algorithm tends to overestimate the COT compared to the transmission method, e.g. between 16.1 and 16.6 UTC. At the end of the presented period between 19.4 and 20 UTC, the lidar signal increases importantly and for this cloud the convergence of our algorithm is less good. Figure 4e shows the value of the cost function after the iteration (normalized by the size of the measurement vector) which indicates the quality of the retrieval. For the above-mentioned period the cost function is large ($\Phi \gg 1$) which means that the algorithm did not converge. This is partly due to the strong attenuation of the signal because of the optically thick cloud that was present during this period, but we will show in the following that this is also related to an insufficient description of the microphysical properties of the ice crystals, in particular the phase function in the backscattering direction.

Our retrievals strongly depend on the phase function in the backscattering direction which defines the backscatter-to-extinction ratio. As explained in Sect. 3.1, we calculate the backscatter-to-extinction ratio for cirrus clouds from Eq. 3 and

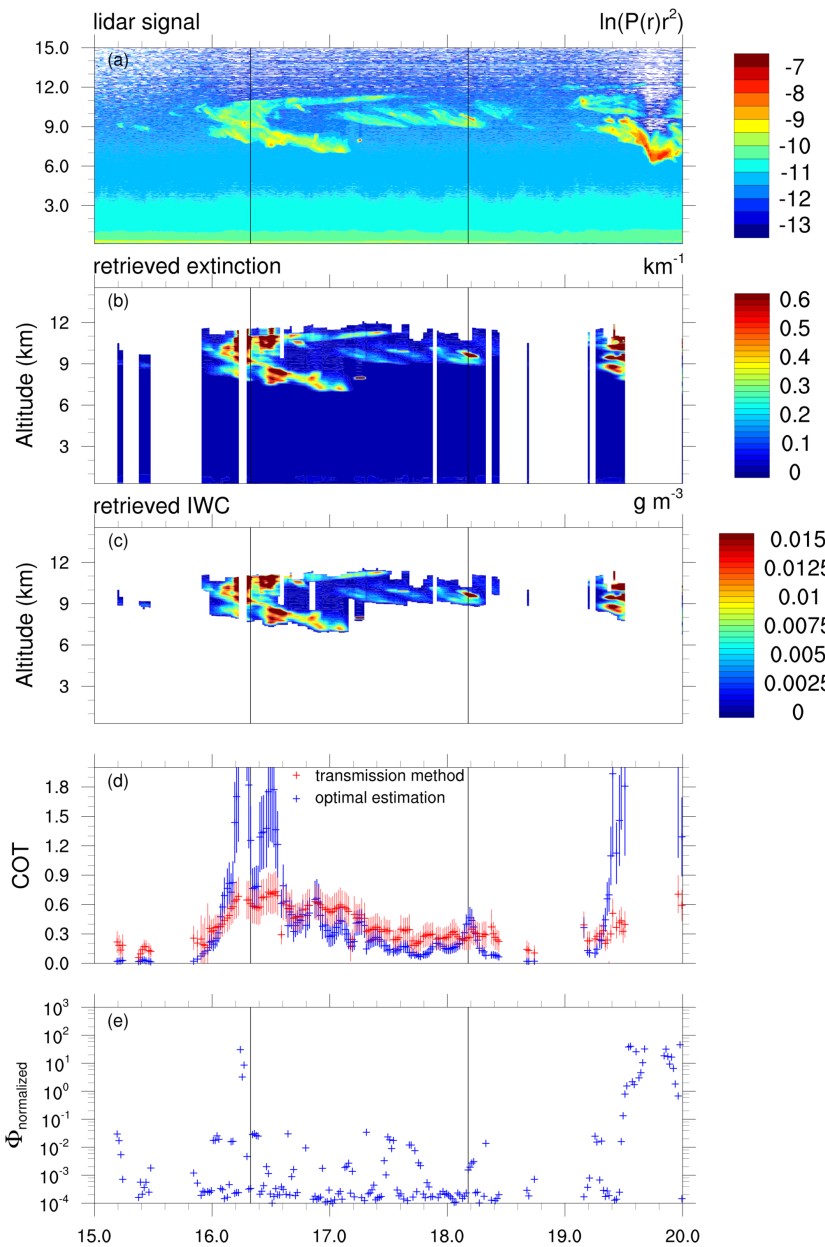

**Figure 4.** Retrieval results for November 30, 2016, 15 to 20 UTC. (a) Logarithm of the measured range-corrected lidar signal. (b) Retrieved extinction profiles (recalculated from the IWC with the BV2015 parametrization inside the cirrus cloud). (c) Retrieved IWC profiles. (d) Cloud Optical Thickness (COT) calculated from the retrieved extinction profile (blue) compared to the COT derived from the transmission method of Young (1995) and Chen et al. (2002) (red). (e) Cost function after the last iteration step normalized by the size of the measurement vector. The vertical lines indicate the profiles at 16.33 UTC and 18.18 UTC which are discussed in more detail.

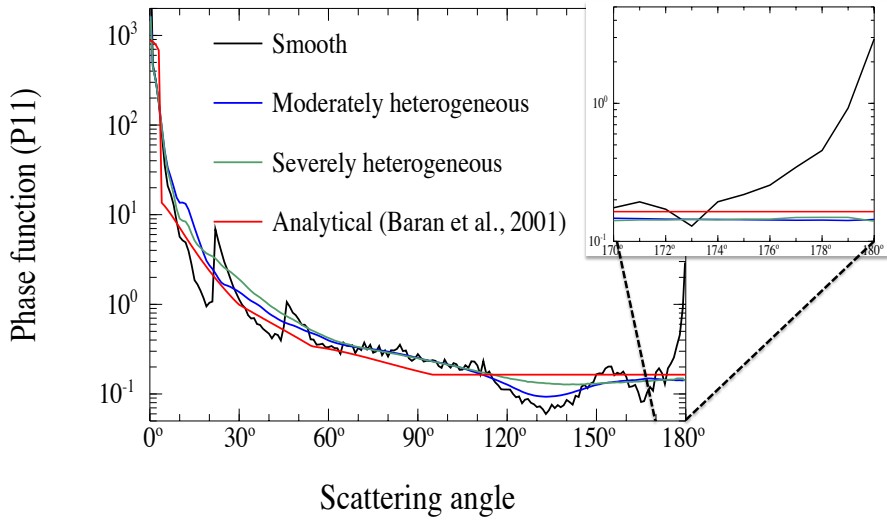

**Figure 5.** Examples of phase functions for different degrees of particle heterogeneity. Black line: phase function for a bulk ice crystal with a smooth surface, blue line: introduction of some heterogeneity, green line: maximum degree of heterogeneity (particle roughness, air bubbles). The red line represents the phase function obtained from the parametrization of Baran et al. (2001).

hence our retrievals strongly depend on the single scattering albedo, $\varpi_0$, and the phase function in the exact backscattering direction, $P_{11}(\pi)$, which are obtained from the BV2015 microphysical model. The single scattering albedo is considered to be represented sufficiently accurately in this model. On the contrary, the phase function, especially in the exact backscattering direction, is much more uncertain.

Figure 5 shows examples of phase functions of ice crystals computed from the ensemble model and the BV2015 parametrization (see Sect. 3.1), for a thin cirrus cloud with a small IWC and a temperature of 250 K. The existence of a backscattering peak strongly depends on the characteristics of the considered particles, especially their heterogeneity represented respectively by their surface roughness and/or by the presence of spherical inclusions (Hess et al., 1998a; C.-Labonnote et al., 2001; Baran and Labonnote, 2006). The black line represents a phase function obtained from the ensemble model, for a bulk ice com-

posed of smooth ice particles (e.g. smooth surface with no heterogeneity), and in this case the phase function shows a strong increase in backscattering direction. Introducing particle heterogeneities, e. g. surface roughness and air bubbles, leads to the disappearance of the backscattering peak (blue and green lines in Fig. 5, computed from the same model and bulk ice but by considering moderately and severely heterogeneous particles, respectively). However, real ice clouds may consist of a mixture of smooth and rough particles and different particle sizes, and their phase functions in the backscattering direction have not

yet been characterized sufficiently accurately. The analytic phase function of Baran et al. (2001) (represented by the red line

in Fig. 5) which is implemented in our algorithm does not include enhanced backscattering. For scattering angles larger than 95° the parametrization assumes a constant value. Recent publications of Zhou and Yang (2015) and Ding et al. (2016) suggest that this assumption is not exact enough to realistically represent the phase function of ice crystals, even for highly heterogeneous particles. They found that a narrow backscattering peak also exists for ice particles with rough surfaces and that the

backscattering is generally underestimated. Zhou and Yang (2015) showed that the phase function of real bulk ice crystals at 180° should be 1.5 to 2.0 times larger than the phase function at 175°, which is clearly not the case for the analytical phase function used in this study.

Having that in mind, we tested the influence of the backscatter-to-extinction ratio in our algorithm. Figure 6 shows the retrieved IWC profiles for the lidar profiles measured on November 30, 2016 at 16.33 UTC (Fig. 6a) and 18.18 UTC (Fig. 6b)

for different backscatter-to-extinction ratios $k'$, where $k' = \kappa \cdot k$ with $k$ the original backscatter-to-extinction ratio computed from Eq. 3. The blue line represents the retrieval result for a factor of $\kappa = 1.0$, the red and green lines for modified backscatter-to-extinction ratios by factors of $\kappa = 1.5$ and $\kappa = 2.0$, respectively. When the backscatter-to-extinction ratio is enhanced, the retrieved IWC decreases evidently. However, the effect of modifying the backscatter-to-extinction ratio is rather strong and integrating the IWC over the whole cloud results for the profile measured at 18.18 UTC in an IWP of $4.22 \pm 1.01 \, \mathrm{g\,m^{-2}}$ for

the backscatter-to-extinction ratio modified by a factor of $\kappa = 2.0$ and $5.98 \pm 1.43 \, \mathrm{g\,m^{-2}}$ for a factor of $\kappa = 1.5$, compared to $10.32 \pm 2.47 \, \mathrm{g\,m^{-2}}$ with the backscatter-to-extinction ratio calculated directly from the BV2015 microphysical model. For the geometrically thick cloud measured at 16.33 UTC, the use of the backscatter-to-extinction ratio computed directly from Eq. 3 results in a strongly increasing IWC towards the cloud top which seems to be unrealistic. This peak of IWC at the cloud top is reduced significantly for retrievals performed with the modified backscatter-to-extinction ratios. Furthermore, the

resulting IWP is reduced by a factor of 4 comparing the retrievals assuming $\kappa = 1.0$ (IWP $= 32.21 \pm 8.93 \, \mathrm{g\,m^{-2}}$) and $\kappa = 2.0$ (IWP $= 8.58 \pm 2.25 \, \mathrm{g\,m^{-2}}$).

As discussed in Sect. 3.1, the challenge of inverting the lidar equation is to find ways to constrain the backscatter-to-extinction ratio which is the major source of uncertainty in the lidar retrieval problem. Stephens et al. (2001) included a visible optical depth in form of an additional measurement in the optimal estimation framework to retrieve both the backscatter-to-

extinction ratio and the extinction profile together. Instead of relying on a retrieval product, such as optical depth, and because the integrated amount of ice depends strongly on the backscatter-to-extinction ratio, we use TIR radiometer measurements to constrain the backscatter-to-extinction ratio of cirrus clouds.

### 3.3.2   Use of TIR radiances to constrain the backscatter-to-extinction ratio

Since TIR radiances are sensitive to the integrated properties of the cloud, in particular the IWP, we can use them to constrain

the amount of ice in the cloud and hence the backscatter-to-extinction ratio. CLIMAT radiometer measurements (see Sect. 2) in its three channels are available on November 30, 2016. To simulate these measurements, the Linearized Discrete Ordinate Radiative Transfer model (LIDORT) (Spurr et al., 2001) has been used. This model requires as inputs profiles of atmospheric temperature, pressure and gases, especially water vapor, which are obtained from ECMWF reanalysis. Furthermore, the optical properties of aerosol and cloud particles deduced from the retrieved IWC and extinction profiles are used in the simulations.

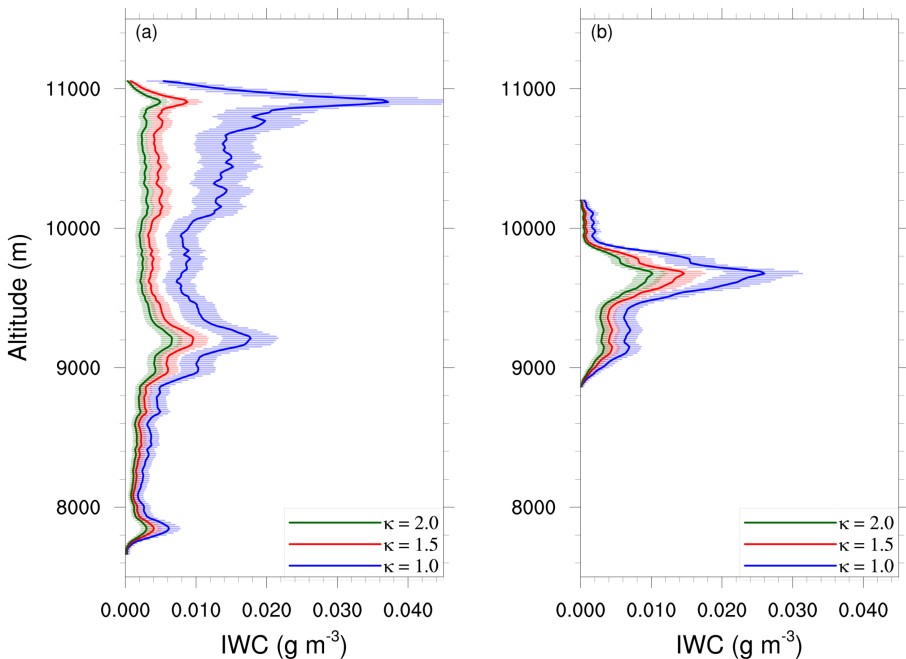

**Figure 6.** Dependence of the retrieved IWC on the backscatter-to-extinction ratio for (a) the lidar profile measured on November 30, 2016 at 16.33 UTC and (b) the lidar profile measured on November 30, 2016 at 18.18 UTC. Shaded zones represent the retrieval errors.

From these profiles, an optical thickness for the lidar wavelength for each model layer is calculated which is linked via Mie theory for aerosols and via the BV2015 parametrization for ice clouds to the optical thickness for the wavelengths in the TIR. Other necessary inputs are the single scattering albedo and the phase function coefficients for the representation of the Legendre polynomial. As mentioned in Sect. 3.1, the aerosol characteristics are obtained from the OPAC database (Hess et al., 1998b)
for the urban aerosol type. Hence, the aerosol model between the lidar and the TIR wavelengths is coherent.

The normalized radiance (emitted by a source of brightness temperature $T_b$) received in channel $C_i$ of the radiometer is characterized by the spectral response $f_i(\lambda)$ of the channel and can be expressed by

$$L_i = \frac{\int_{\Delta\lambda_i} B_\lambda(T_b) f_i(\lambda) \mathrm{d}\lambda}{\int_{\Delta\lambda_i} f_i(\lambda) \mathrm{d}\lambda}, \tag{24}$$

where $\lambda$ is the wavelength, $B_\lambda(T_b)$ the Planck function and $\Delta\lambda_i$ the spectral bandpass of channel $C_i$.
Figure 7 shows the normalized radiances measured with CLIMAT in its three channels on November 30, 2016, between 15 and 20 UTC. The black lines represent the simulation with LIDORT for a cloud-free atmosphere taking into account the aerosol extinction in the layers below the cloud deduced from the lidar measurements and considering water vapor and temperature profiles from an ECMWF reanalysis at 12 UTC. All three channels show an increase of the signal due to the clouds present between 16 and 18.5 UTC as well as after 19.2 UTC. However, the signal for the first cloudy period is much smaller than for
the second period because of the smaller COT in combination with a higher and hence colder cloud base altitude compared to

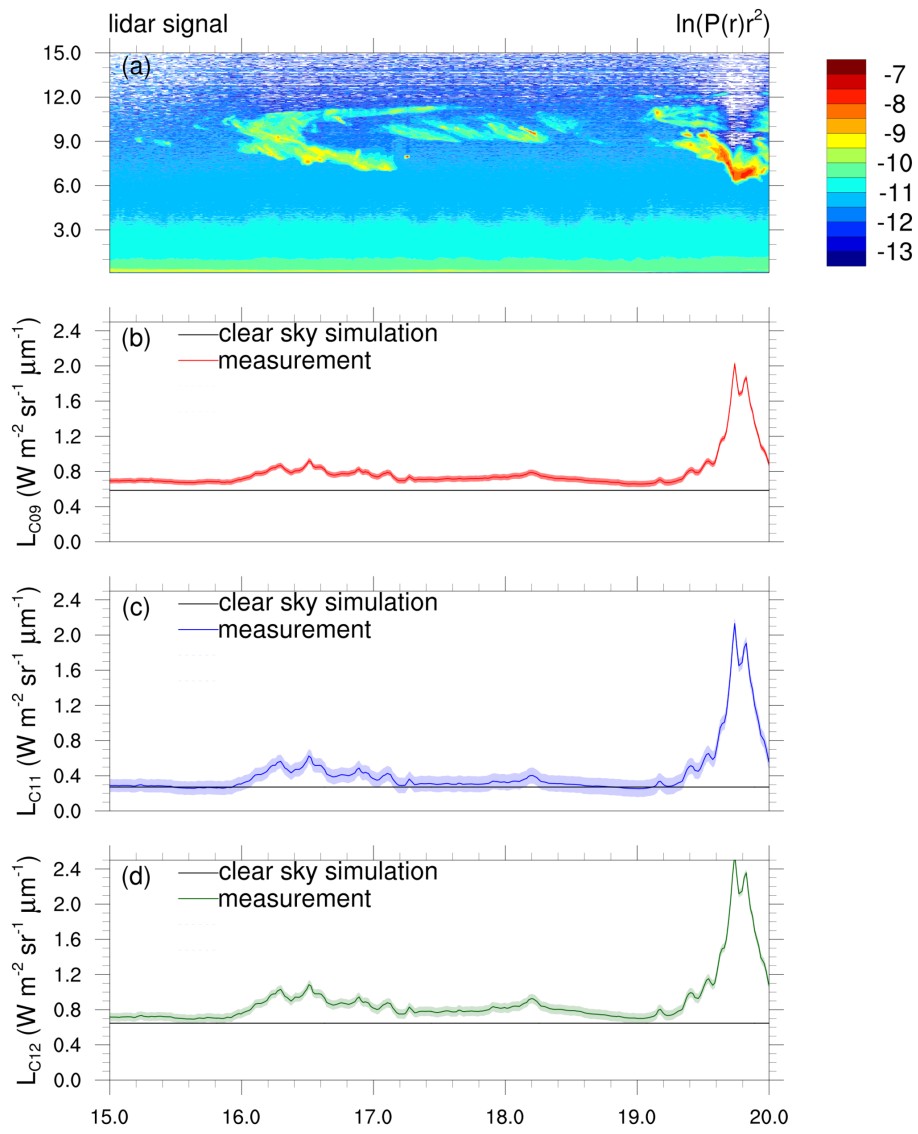

**Figure 7.** (a) Logarithm of the measured range-corrected lidar signal for November 30, 2016, 15 to 20 UTC. (b) TIR radiometer measurements for channel C09, (c) for channel C11, and (d) for channel C12. Shaded zones represent the error range of the measurement. The black lines in (b) to (d) represent the simulation with LIDORT for the three channels under cloud-free conditions taking into account the aerosol extinction in the lowest layers obtained from the lidar and water vapor and temperature profiles from an ECMWF reanalysis at 12 UTC.

the second period. Furthermore, Fig. 7 shows that the simulated radiances for the cloud-free atmosphere are within the error-range of the measured radiances under cloud-free conditions (between 15 and 16 UTC as well as around 19 UTC) for channels C11 and C12. On the contrary, the measured radiance for channel C09 is not reproduced by the radiative transfer simulations which may be due to an insufficient knowledge of the spectral response function ($f_i(\lambda)$ in Eq. 24) for this channel resulting in

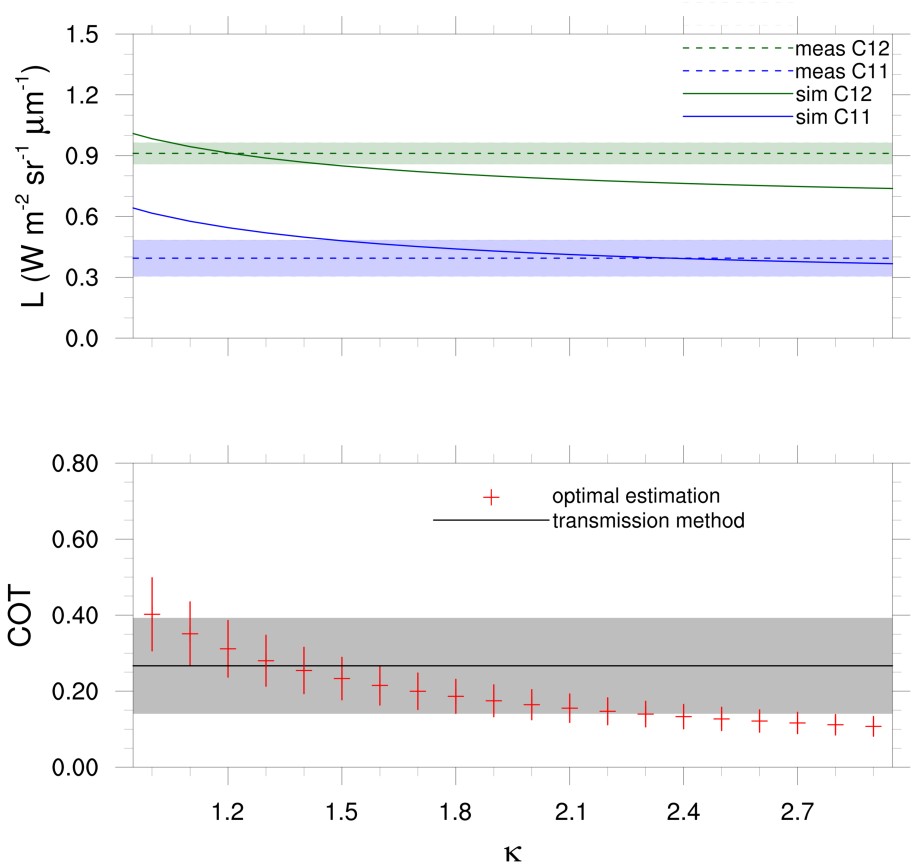

**Figure 8.** (a) Dependence of the simulated normalized radiances on a factor for the backscatter-to-extinction ratio ranging from 1.0 to 3.0 for the lidar profile measured on November 30, 2016 at 18.18 UTC. The CLIMAT measurements of channels C11 and C12 are represented by the dashed lines, shaded zones indicate the measurement error. (b) Corresponding COT. The black line represents the COT derived from the transmission method and the shaded grey zone its error range.

a convergence issue of our retrieval algorithm. We therefore decided to not take this channel into account for the remainder of this paper.

The idea of our method is to use these TIR radiances to constrain the backscatter-to-extinction ratio correction factor ($\kappa$) and hence the phase function in the backscattering direction. In the following, we aim to show the potential of such an approach. Figure 8a shows the simulated normalized radiances (in $\mathrm{W\,m^{-2}\,sr^{-1}\,\mu m^{-1}}$) for the lidar profile measured on November 30, 2016 at 18.18 UTC as a function of the correction factor $\kappa$ for the two CLIMAT channels C11 (represented in blue) and C12 (green). The dashed lines indicate the measurements and the shaded zones around it the measurement error. Figure 8b presents the corresponding COT computed from the retrieved extinction profile (red crosses), while the black line represents the COT derived from the transmission method with its according error range (shaded grey zone). With increasing correction

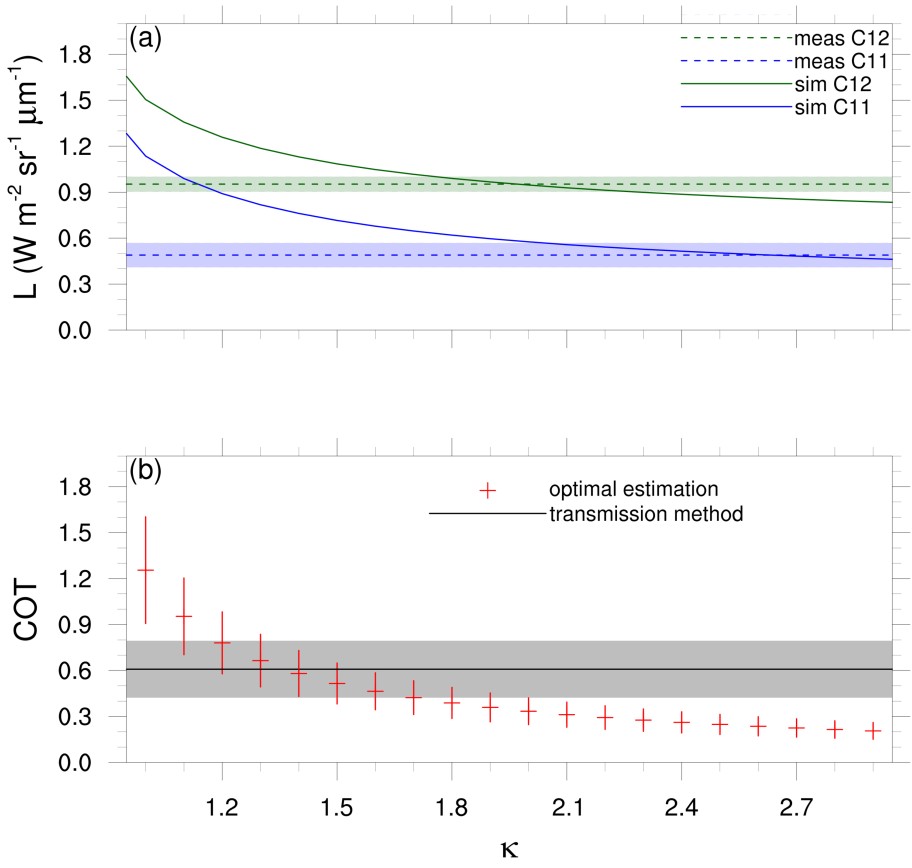

**Figure 9.** Same as Fig. 8 for the lidar profile measured on November 30, 2016 at 16.33 UTC.

factor for the backscatter-to-extinction ratio, the retrieved COT decreases which causes the simulated radiances to decrease as well. Furthermore, one can see from Fig. 8a that the simulated radiances for channel C12 are within the error range of the measurements for a correction factor between $\kappa = 1.1$ and $\kappa = 1.5$ whereas for channel C11 this is the case for correction factors larger than $\kappa = 1.4$. This leads to the conclusion that for this profile the correction factor for the backscatter-to-extinction

5   ratio should range between 1.4 and 1.5 to find a retrieval of the IWC profile that would allow both lidar and TIR forward model to converge towards the corresponding measurements. Additionally, the COT computed from the retrieved extinction profile (see Sect. 3.3.1) agrees well with the COT derived from the transmission method for this range of correction factors (see Fig. 8b). These results indicate that the TIR radiances can help to refine the phase function in the backscattering direction and that the analytic phase function of Baran et al. (2001) may not be exact enough to represent the phase function of real ice crystals

10   in the exact backscattering direction.

Figure 9 shows the same analysis for the lidar profile measured at 16.33 UTC. The COT from the optimal estimation method largely overestimates the COT from the transmission method for this profile when the retrieval is performed with a correction factor of $\kappa = 1.0$. Moreover, Fig. 6a showed a rather unrealistic increase of IWC at the cloud top in this case. Figure 9a suggests that the correction factor constrained by the TIR radiances should range between $\kappa = 2.0$ and $\kappa = 2.3$. A correction factor from this interval reduces the retrieved COT considerably and slightly underestimates the COT obtained from the transmission method.

However, the curves shown in Figs. 8 and 9 underline the importance of the quality of the measurements in the TIR since small changes in the radiances may lead to very different retrieved microphysics. Furthermore, Dubuisson et al. (2008) showed that the atmosphere, especially the water vapor, has a very important influence on ground-based TIR radiometer measurements and that the sensitivity of these measurements to cloud properties is weaker for moist atmospheres. However, the ECMWF reanalysis profile of water vapor indicates a rather dry atmosphere for this day with a total amount of $0.62\,\mathrm{g\,m}^{-2}$ in the atmospheric column, so the water vapor as well as the low aerosol optical depth have a rather small influence on the TIR radiances measured during this case study.

The results presented in this section show that the ensemble of measurements should be used to find a retrieval that corresponds best to all available information. As mentioned above, the optimal estimation method is a well-adapted tool to use different kinds of measurements in a common retrieval framework. The results shown here confirm that the TIR radiances provide an additional constraint for the amount of ice inside the cloud which strongly depends on the backscatter-to-extinction ratio. Therefore, the phase function in the backscattering direction can be constrained by the TIR radiometer measurements under the assumption that the single scattering albedo is known sufficiently accurately (see Eq. 3). As a consequence, we included the TIR radiometer measurements in the optimal estimation framework of the lidar only algorithm to retrieve in addition to the IWC/extinction profiles the correction factor $\kappa$ for the phase function in the backscattering direction. This newly developed synergistic algorithm is presented in the following section.

## 4 Synergy algorithm lidar - TIR

### 4.1 Integration of the TIR radiances in the optimal estimation framework

The synergy algorithm is an expansion of the lidar only algorithm which integrates the TIR radiometer measurements in the optimal estimation method. The new state vector contains in addition to the elements of the previous state vector given by Eq. 9 the correction factor $\kappa$ for the phase function in the exact backscattering direction:

$$\boldsymbol{x} = [\sigma_p(r_1), \sigma_p(r_2), \ldots, \sigma_p(r_{j\_\mathrm{bot}-1}), \mathrm{IWC}(r_{j\_\mathrm{bot}}), \ldots, \mathrm{IWC}(r_{j\_\mathrm{top}}), \sigma_p(r_{j\_\mathrm{top}+1}), \ldots, \sigma_p(r_N), \kappa]^T, \tag{25}$$

where the new phase function $P'_{11}(\pi)$ in backscattering direction is related to the previous one via $P'_{11}(\pi) = \kappa \cdot P_{11}(\pi)$.

The measurement vector, initially containing the logarithm of the calibrated range-corrected lidar signal, is expanded by the measured radiances from the two channels of the CLIMAT instrument discussed above:

$$\boldsymbol{y} = [\ln(C \cdot P(r_1)r_1^2), \ln(C \cdot P(r_2)r_2^2), \ldots, \ln(C \cdot P(r_N)r_N^2), L_{\mathrm{C11}}, L_{\mathrm{C12}}]^T. \tag{26}$$

The forward model for the lidar is the same as in case of the lidar only algorithm, given by the lidar equation in form of Eq. 11, with the only modification that the backscatter-to-extinction ratio for ice cloud layers is now calculated by

$$k' = \varpi_0 \cdot P'_{11}(\pi) = \varpi_0 \cdot \kappa \cdot P_{11}(\pi). \tag{27}$$

The forward model for the TIR radiances is the above-mentioned radiative transfer model LIDORT (Spurr et al., 2001). The advantage of this model is that it provides not only radiances but also weighting functions for atmospheric and surface parameters. That means the Jacobians for surface parameters such as emissivity or temperature, profiles of Jacobians for the temperature, atmospheric gas or IWC profiles, as well as column Jacobians for the integrated quantities, for example the (Jacobian about the) integrated water vapor in the whole atmospheric column, can be obtained together with the radiances from one single simulation. Therefore the use of this model considerably reduces the computation time of the algorithm in comparison to finite difference calculations to obtain Jacobians. This numerical efficiency allows the use of a fine vertical resolution in the radiative transfer calculations without exceeding reasonable computation times. Hence, the radiative transfer calculations can be realized on a vertical grid corresponding to the lidar resolution inside the cirrus cloud (outside the cloud the vertical resolution is defined by the ECMWF reanalysis profiles on 137 levels). That means for the radiative transfer calculations in our algorithm, the extinction profile inside the cirrus cloud is calculated with the BV2015 microphysical model from the IWC given on the lidar resolution. As a consequence, thanks to the synergy with the lidar measurements and to the numerical efficiency of the radiative transfer model, our algorithm does not have to assume a homogeneous cloud like most inversion algorithms for cloud properties do.

The Jacobian of the synergistic algorithm contains in addition to the Jacobian of the lidar only algorithm two new rows for the sensitivity of the TIR forward model to each state vector parameter and one new column for the sensitivity of the forward model to the new state vector element $\kappa$:

$$\mathbf{K} = \begin{pmatrix}
\frac{\partial F_1}{\partial \sigma_{p,1}} & \cdots & \frac{\partial F_1}{\partial \sigma_{p,j\_bot-1}} & \frac{\partial F_1}{\partial IWC_{j\_bot}} & \cdots & \frac{\partial F_1}{\partial IWC_{j\_top}} & \frac{\partial F_1}{\partial \sigma_{p,j\_top+1}} & \cdots & \frac{\partial F_1}{\partial \sigma_{p,N}} & \frac{\partial F_1}{\partial \kappa} \\
\frac{\partial F_2}{\partial \sigma_{p,1}} & \cdots & \frac{\partial F_2}{\partial \sigma_{p,j\_bot-1}} & \frac{\partial F_2}{\partial IWC_{j\_bot}} & \cdots & \frac{\partial F_2}{\partial IWC_{j\_top}} & \frac{\partial F_2}{\partial \sigma_{p,j\_top+1}} & \cdots & \frac{\partial F_2}{\partial \sigma_{p,N}} & \frac{\partial F_2}{\partial \kappa} \\
\vdots & \cdots & \vdots & \vdots & \cdots & \vdots & \vdots & \cdots & \vdots & \vdots \\
\frac{\partial F_{j\_bot-1}}{\partial \sigma_{p,1}} & \cdots & \frac{\partial F_{j\_bot-1}}{\partial \sigma_{p,j\_bot-1}} & \frac{\partial F_{j\_bot-1}}{\partial IWC_{j\_bot}} & \cdots & \frac{\partial F_{j\_bot-1}}{\partial IWC_{j\_top}} & \frac{\partial F_{j\_bot-1}}{\partial \sigma_{p,j\_top+1}} & \cdots & \frac{\partial F_{j\_bot-1}}{\partial \sigma_{p,N}} & \frac{\partial F_{j\_bot-1}}{\partial \kappa} \\
\frac{\partial F_{j\_bot}}{\partial \sigma_{p,1}} & \cdots & \frac{\partial F_{j\_bot}}{\partial \sigma_{p,j\_bot-1}} & \frac{\partial F_{j\_bot}}{\partial IWC_{j\_bot}} & \cdots & \frac{\partial F_{j\_bot}}{\partial IWC_{j\_top}} & \frac{\partial F_{j\_bot}}{\partial \sigma_{p,j\_top+1}} & \cdots & \frac{\partial F_{j\_bot}}{\partial \sigma_{p,N}} & \frac{\partial F_{j\_bot}}{\partial \kappa} \\
\vdots & \cdots & \vdots & \vdots & \cdots & \vdots & \vdots & \cdots & \vdots & \vdots \\
\frac{\partial F_{j\_top}}{\partial \sigma_{p,1}} & \cdots & \frac{\partial F_{j\_top}}{\partial \sigma_{p,j\_bot-1}} & \frac{\partial F_{j\_top}}{\partial IWC_{j\_bot}} & \cdots & \frac{\partial F_{j\_top}}{\partial IWC_{j\_top}} & \frac{\partial F_{j\_top}}{\partial \sigma_{p,j\_top+1}} & \cdots & \frac{\partial F_{j\_top}}{\partial \sigma_{p,N}} & \frac{\partial F_{j\_top}}{\partial \kappa} \\
\frac{\partial F_{j\_top+1}}{\partial \sigma_{p,1}} & \cdots & \frac{\partial F_{j\_top+1}}{\partial \sigma_{p,j\_bot-1}} & \frac{\partial F_{j\_top+1}}{\partial IWC_{j\_bot}} & \cdots & \frac{\partial F_{j\_top+1}}{\partial IWC_{j\_top}} & \frac{\partial F_{j\_top+1}}{\partial \sigma_{p,j\_top+1}} & \cdots & \frac{\partial F_{j\_top+1}}{\partial \sigma_{p,N}} & \frac{\partial F_{j\_top+1}}{\partial \kappa} \\
\vdots & \cdots & \vdots & \vdots & \cdots & \vdots & \vdots & \cdots & \vdots & \vdots \\
\frac{\partial F_N}{\partial \sigma_{p,1}} & \cdots & \frac{\partial F_N}{\partial \sigma_{p,j\_bot-1}} & \frac{\partial F_N}{\partial IWC_{j\_bot}} & \cdots & \frac{\partial F_N}{\partial IWC_{j\_top}} & \frac{\partial F_N}{\partial \sigma_{p,j\_top+1}} & \cdots & \frac{\partial F_N}{\partial \sigma_{p,N}} & \frac{\partial F_N}{\partial \kappa} \\
\frac{\partial F_{C11}}{\partial \sigma_{p,1}} & \cdots & \frac{\partial F_{C11}}{\partial \sigma_{p,j\_bot-1}} & \frac{\partial F_{C11}}{\partial IWC_{j\_bot}} & \cdots & \frac{\partial F_{C11}}{\partial IWC_{j\_top}} & \frac{\partial F_{C11}}{\partial \sigma_{p,j\_top+1}} & \cdots & \frac{\partial F_{C11}}{\partial \sigma_{p,N}} & \frac{\partial F_{C11}}{\partial \kappa} \\
\frac{\partial F_{C12}}{\partial \sigma_{p,1}} & \cdots & \frac{\partial F_{C12}}{\partial \sigma_{p,j\_bot-1}} & \frac{\partial F_{C12}}{\partial IWC_{j\_bot}} & \cdots & \frac{\partial F_{C12}}{\partial IWC_{j\_top}} & \frac{\partial F_{C12}}{\partial \sigma_{p,j\_top+1}} & \cdots & \frac{\partial F_{C12}}{\partial \sigma_{p,N}} & \frac{\partial F_{C12}}{\partial \kappa}
\end{pmatrix}. \tag{28}$$

The sensitivities of the TIR radiances to the extinction profile outside the cloud are set to zero since they are assumed to be small. The correction factor $\kappa$ does not have a direct influence on the TIR radiances, thus the last two elements in the last column of the Jacobian matrix (**K**) are also set to zero (see Eq. 28). The sensitivities of the TIR radiances to the IWC profile inside the cloud are calculated directly in LIDORT. Finally, the partial derivatives of the lidar forward model with respect to $\kappa$

are set to zero outside the cloud and calculated analytically as derivation of the forward model (Eq. 11) for the ice cloud layers,

$$\frac{\partial F_j}{\partial \kappa} = \frac{\varpi_0 \cdot P_{11}(\pi) \cdot \sigma_{p,j}}{\beta_{m,j} + \varpi_0 \cdot P_{11}(\pi) \cdot \kappa \cdot \sigma_{p,j}}, \tag{29}$$

where the layer extinction $\sigma_{p,j}$ is calculated from the IWC with the BV2015 parametrization.

As in the lidar only algorithm, the variance-covariance matrices in the synergy algorithm are also considered to be diagonal.

Concerning the lidar, they are defined in the same way as in the lidar only algorithm (see Sect. 3.2) with the only difference that the error for the backscatter-to-extinction ratio in ice cloud layers is no longer considered since we retrieve with the new algorithm a correction factor for the phase function in the backscattering direction which is directly related to the backscatter-to-extinction ratio. Instead, an error of 1% on the single scattering albedo is integrated (compare with Eq. 27). For the variance-covariance matrix of the TIR forward model the considered non-retrieved parameters (and the errors attributed to them) are the

following: surface emissivity (2%), surface temperature (1 K) and the profiles of atmospheric temperature (1 K for each layer), water vapor (10% for each layer) and ozone (2% for each layer). The standard deviations are calculated via

$$\sigma_{b_j} = \frac{\partial F}{\partial b_j} \cdot b_j \cdot \frac{p_{b_j}(\%)}{100}, \tag{30}$$

where $b_j$ represents the considered non-retrieved parameter, $p_{b_j}(\%)$ its error in percent and $\frac{\partial F}{\partial b_j}$ the sensitivity of the forward model to this parameter. As mentioned above, the latter can be calculated directly in LIDORT for all desired parameters (for

a detailed description of the calculation of Jacobians in LIDORT the reader is referred to the LIDORT User's Guide, (Spurr, 2012)). The elements of the diagonal variance-covariance matrix are then given by

$$S_{\epsilon,ii_{\text{TIR}}} = \sigma_{y,i_{\text{TIR}}}^2 + \sum_j \sigma_{b_j,i_{\text{TIR}}}^2, \tag{31}$$

where $\sigma_{y,i_{\text{TIR}}}^2$ represents the measurement errors for each of the two channels of the TIR radiometer. This error depends on the calibration procedure and on the temperature of the instrument during the measurement because its sensitivity is a function

of temperature. The calibration of the instrument is performed in the laboratory at room temperature. Unfortunately, our instrument does not have a thermal enclosure system and during field measurements it is exposed to atmospheric temperature influences. On November 30, 2016, the atmospheric temperature was low. Due to the poor knowledge of a coefficient to correct for the instrument's temperature, the assumed errors on the measured radiances are rather large. Especially, for channel C11 the error arising from this temperature correction ranges between 15 and 30 % depending on the value of the radiance.

The largest error percentages occur for small normalized radiances in combination with a cold instrument temperature. The radiances measured by channel C12 are larger and hence the error on the measurements of this channel is smaller and ranges between 5 and 8 %.

**Table 1.** TIR forward model and measured normalized radiances ($\mathrm{W\,m^{-2}\,sr^{-1}\,\mu m^{-1}}$) for November 30, 2016 at 18.18 UTC.

|  | C11 | C12 |
|---|---|---|
| TIR forward model of the a priori | 0.3173 | 0.6889 |
| TIR forward model after convergence | $0.4853 \pm 0.0209$ | $0.8548 \pm 0.0393$ |
| Measurement | $0.3885 \pm 0.0897$ | $0.9054 \pm 0.0534$ |

## 4.2 Preliminary results

This section presents some preliminary results of our new algorithm. Figure 10 shows the same example as given in Fig. 2 but obtained from the synergy algorithm. The a priori used for the extinction/IWC profiles in the synergy algorithm is the same as in the lidar only algorithm (see Sect. 3.3). Figure 10b shows that once the algorithm converged, the lidar forward model and

the measured lidar signal overlay each other almost perfectly. As in the lidar only algorithm, the relative difference between the forward model and the measurement is smaller than 1% for all layers. Thus, the good convergence found in the lidar only algorithm is confirmed in the synergy algorithm. Table 1 summarizes the radiometer measurements and the TIR forward model (LIDORT) corresponding to this profile before and after the algorithm's convergence (expressed in normalized radiances in $\mathrm{W\,m^{-2}\,sr^{-1}\,\mu m^{-1}}$). Since the values of the TIR forward model after the iteration process are within the error range of the

measurements, it can be concluded that the algorithm converged in the TIR as well.

The retrieved value for the correction factor $\kappa$ for the phase function in the backscattering direction is $1.48 \pm 0.33$ which is close to the range of 1.5 to 2.0 found in the literature (Zhou and Yang, 2015) and confirms the result shown in Fig. 8. The corresponding retrieved IWC and extinction profiles are shown in Fig. 11. By comparing them to the result of the lidar only algorithm (Fig. 3), it is obvious that the IWC and the extinction are smaller for the synergy algorithm because the backscatter-to-extinction

ratio in the ice cloud is larger. The resulting IWP is $6.13 \pm 2.19\,\mathrm{g\,m^{-2}}$ compared to the initial IWP of $10.32 \pm 2.47\,\mathrm{g\,m^{-2}}$ from the lidar only algorithm. As a consequence, the COT is also considerably reduced to a value of $0.239 \pm 0.085$ compared to $0.402 \pm 0.096$ from the lidar only algorithm and is in good agreement with the COT of $0.267 \pm 0.126$ derived from the transmission method considering the error ranges. This indicates that the result of the synergy algorithm is more coherent than the result of the lidar only algorithm because the backscatter-to-extinction ratio has been characterized more realistically.

The application of the synergy algorithm to the profile measured at 16.33 UTC on November 30, 2016 results in a factor $\kappa$ of $2.15 \pm 0.33$. Similarly as for the profile at 18.18 UTC, this value for the correction factor corresponds to the region where the simulated TIR radiances converge to the measurements in Fig. 9. Table 2 shows the radiometer measurements and the TIR forward model after the iteration. As for the profile at 18.18 UTC, the values of the TIR forward model after the iteration process are within the error range of the measurements. Thus, it can be concluded that the algorithm found a

solution allowing both the lidar and the TIR forward model to converge towards the corresponding measurements. For the retrieved correction factor, the large IWC peak at the cloud top observed from the lidar only algorithm for a correction factor of $\kappa = 1.0$ is considerably reduced which results in a more realistic shape of the IWC profile. The IWP obtained from the

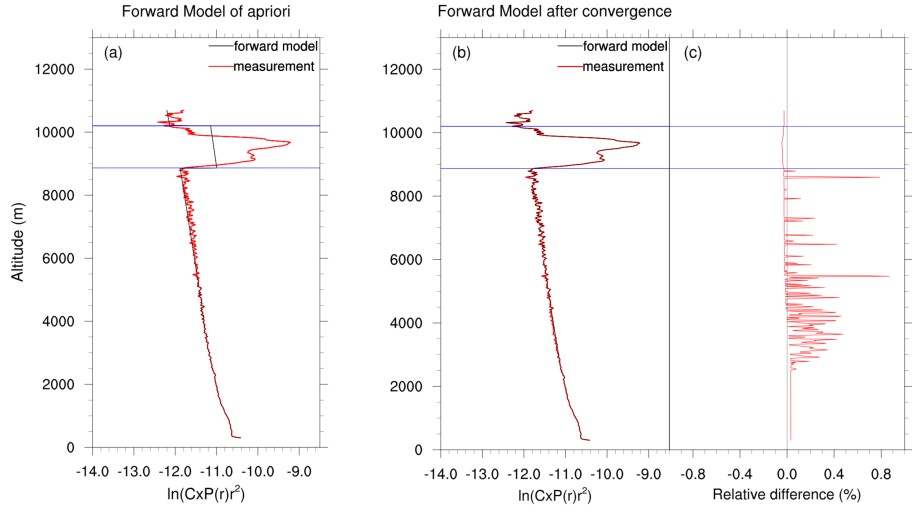

**Figure 10.** Lidar forward model of the (a) a priori, and (b) after the last iteration step (represented by the black lines) from the synergy algorithm for the lidar profile measured on November 30, 2016 at 18.18 UTC. The red lines represent the measurement, the horizontal blue lines indicate the defined cloud base and top altitudes. (c) Relative difference between the forward model and the measurement after the iteration.

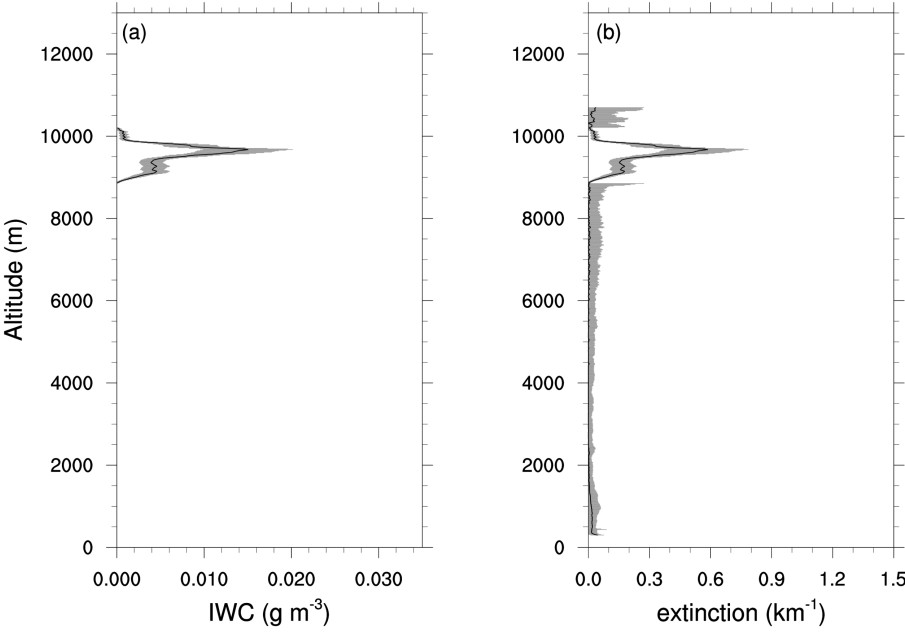

**Figure 11.** (a) Retrieved IWC profile, and (b) retrieved extinction profile from the synergy algorithm for the lidar profile measured on November 30, 2016 at 18.18 UTC. Shaded areas represent the total error on the retrieved quantities.

**Table 2.** TIR forward model and measured normalized radiances (W m$^{-2}$ sr$^{-1}$ μm$^{-1}$) for November 30, 2016 at 16.33 UTC.

|                                        | C11                 | C12                 |
| -------------------------------------- | ------------------- | ------------------- |
| TIR forward model of the a priori      | 0.4012              | 0.7730              |
| TIR forward model after convergence    | $0.5508 \pm 0.0214$ | $0.9220 \pm 0.0395$ |
| Measurement                            | $0.4925 \pm 0.0793$ | $0.9567 \pm 0.0492$ |

synergy algorithm is $7.79 \pm 2.54$ g m$^{-2}$ compared to the above-mentioned value of $32.21 \pm 8.93$ g m$^{-2}$ from the lidar only algorithm. Hence, the retrieval with the synergy algorithm results in an important decrease of the IWP. However, the COT of $0.304 \pm 0.099$ obtained from the synergy algorithm underestimates the COT of $0.608 \pm 0.186$ obtained from the transmission method.

Finally, Fig. 12 presents the temporal evolution of the retrieval results from the synergy algorithm for the time period from 15 to 20 UTC on November 30, 2016. Figure 12a reiterates the measured lidar signal already shown in Fig. 4a, Fig. 12b shows the retrieved IWC profiles, and Fig. 12c the retrieved correction factor $\kappa$ for the phase function in the backscattering direction (blue) and the according lidar ratio in sr (red) which might be easier to interpret. Figure 12d shows the comparison of the COT obtained from the synergy algorithm (blue) and the transmission method (red) and Fig. 12e presents the TIR radiometer

measurements for channels C12 and C11 in green and blue, respectively, and the forward model after convergence for channel C12 in red and for channel C11 in violet, including their uncertainties. This plot indicates that the majority of retrievals converges well in the TIR. Furthermore, retrieval results are only shown in the other plots of this panel if the normalized cost function is much smaller than unity. Hence, the large number of results shown in this figure also indicates the overall good convergence of our algorithm. The only retrievals that did not converge correspond to either optically very thin clouds or to

clouds which are thick enough to attenuate the lidar signal completely (e.g. around 19.8 UTC). In the second case, this could be related to the reduced size of the Jacobian mentioned in Sect. 3.2. However, we believe that this non-convergence is more likely due to physical reasons since the cloud base at this time was located in low altitudes (around 6 km) and the temperature in this altitude was rather warm for a cirrus cloud (between 245 and 250 K). In this temperature range, the presence of supercooled liquid droplets is possible which is not included in the BV2015 microphysical model. Hence, this model does probably not

represent the optical properties of this cloud accurately enough. Nevertheless, compared to the lidar only algorithm the synergy algorithm converged for more profiles between 19.4 and 20 UTC and the retrieved COT compares well to the COT from the transmission method during this period. Hence, the TIR helped to constrain the backscatter-to-extinction ratio through the IWP, allowing a better coherence between the visible and the TIR forward model.

However, between 16 and 18.3 UTC the COT obtained from the synergy algorithm underestimates the COT derived from

the transmission method for most of the retrievals (except around 18.2 UTC). It should be noted that the COT obtained from the transmission method is an effective COT$^\star$ and that the real COT depends on the multiple scattering factor where COT$^\star = \eta \cdot$ COT. Hence, for Fig. 12c, the effective optical thickness obtained from the transmission method has been divided by the assumed multiple scattering factor for ice clouds ($\eta = 0.75$) in order to be consistent with the retrievals from the synergy

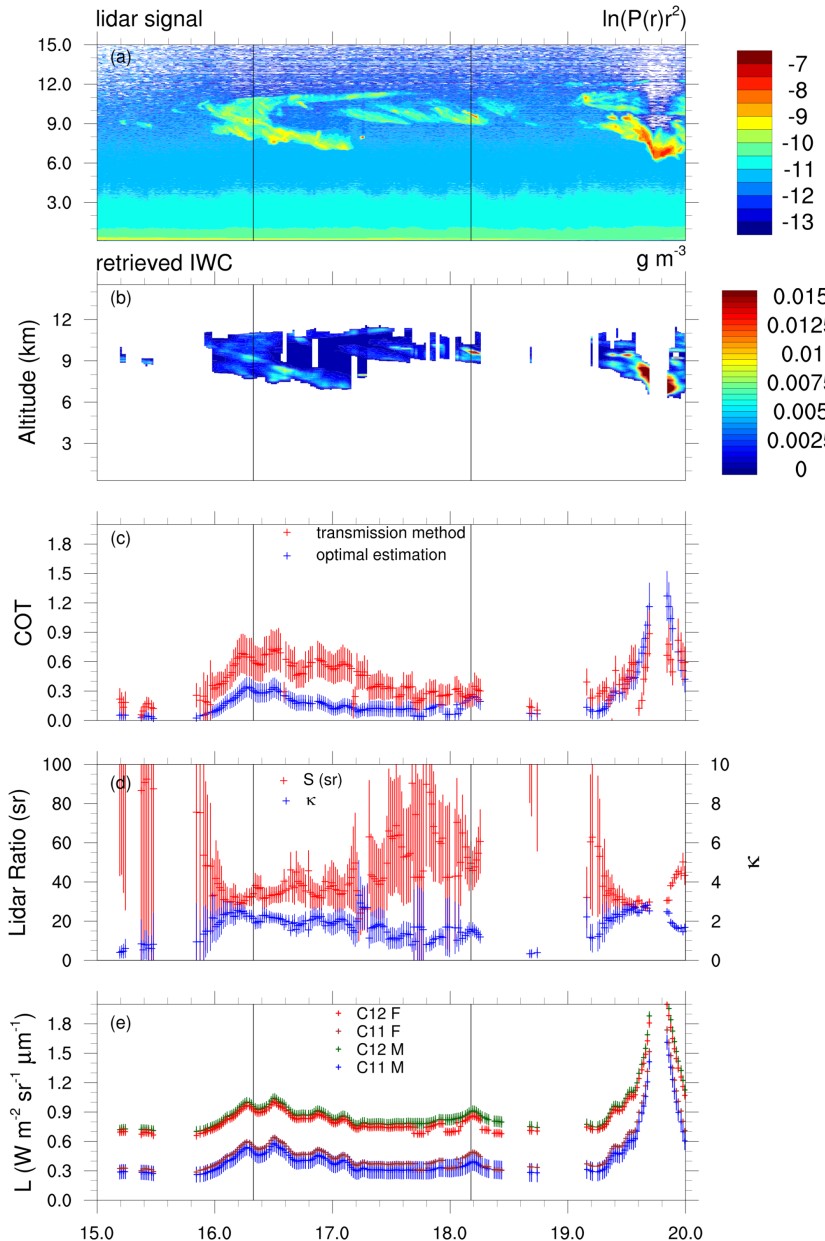

**Figure 12.** Retrieval results of the synergy algorithm for November 30, 2016, 15 to 20 UTC. (a) Logarithm of the measured range-corrected lidar signal. (b) Retrieved IWC profiles. (c) Retrieved factor $\kappa$ (right axis, represented by blue crosses) and the corresponding lidar ratio in sr (left axis, represented by red crosses). (d) COT from synergy algorithm (blue) and from transmission method (red). (e) Thermal infrared radiometer measurements (C11 M and C12 M) and the converged forward model results (C11 F and C12 F), expressed as normalized radiances.

algorithm. Thus, this corrected COT depends strongly on the assumed multiple scattering factor. Applying a larger value of $\eta$ (and hence reduce the effect of multiple scattering) would reduce the optical thickness obtained from this method and would result in a value which is closer to the result from the synergy algorithm. On the contrary, the COT retrieved with the synergy algorithm is constrained by the TIR radiometer measurements and remains constant when applying another multiple scattering factor. In the synergy algorithm, the retrieval of the correction factor $\kappa$ for the phase function in the backscattering direction would change and hence the microphysics of the cirrus cloud. The influence of the multiple scattering factor on the retrievals of our synergy algorithm has to be further investigated in future studies in order to draw more sophisticated conclusions and the retrievals shown here should be understood as a first test to show the potential of the algorithm.

However, the multiple scattering factor alone cannot explain the inconsistency between the COT retrieved with the synergy algorithm and the COT derived from the transmission method. Another possible reason for this discrepancy may arise from the uncertainty in the transmission method itself because it depends on a good characterization of the molecular signal above the cloud and a good estimation of the cloud top altitude. These parameters are related with rather large uncertainties due to the quite noisy micro-pulse lidar signal in the high altitudes of cirrus clouds. Furthermore, the discrepancy between the two COTs could also originate from a potential bias in the TIR radiometer measurements due to an inaccurate temperature correction as mentioned in Sect. 4.1, or from a potential bias in the TIR forward model due to an inaccurate description of the atmospheric water vapor profile since the TIR radiometer measurements are very sensitive to water vapor (Dubuisson et al., 2008). Finally, the difference in the COTs from the synergy algorithm and the transmission method could also originate from the microphysical model which might not be perfect. The extinction at the lidar wavelength, which is calculated based on the IWP constrained by the TIR radiometer measurements, could be slightly underestimated. Figure 12c shows that if the IWP is larger, the difference between the COTs from the two methods becomes smaller. This can be explained by the fact that the contribution of the water vapor in the TIR radiometer measurements is more important for thin clouds than for thick clouds leading to an underestimation of the IWP and consequently an underestimation of the extinction, especially in case of thin cirrus clouds.

Despite of these limitations, the retrievals of the lidar ratio shown in Fig. 12d are promising. For the geometrically and optically thicker cloud between 16 and 17.2 UTC where a considerable increase in the measured TIR radiances was observed (see Fig. 7), the average value of the retrieved lidar ratio is $35.6 \pm 4.5\,\mathrm{sr}$ which is in agreement with the lidar ratios for cirrus clouds reported in the literature ranging between 20 and 40 sr (e.g., Chen et al., 2002; Giannakaki et al., 2007; Josset et al., 2012; Garnier et al., 2015). Between 17.2 and 18.2 UTC the retrieved lidar ratios are much higher (on average $52.2 \pm 17.0\,\mathrm{sr}$) but the cloud observed during this period is optically very thin and the signal in the TIR radiances very small, so it is not surprising that our algorithm reaches its limit here. For the optically thicker cloud between 19.2 and 20 UTC the average of the retrieved lidar ratio is $35.8 \pm 9.1\,\mathrm{sr}$, corresponding to the literature again. However, as discussed above, the retrievals of the correction factor $\kappa$ and thus the lidar ratio depend on the assumed multiple scattering factor.

Nevertheless, this new synergistic algorithm suggests that using information from both, active in the visible part of the electromagnetic spectrum and passive in the TIR part, allows to obtain new information on bulk ice optical properties, especially on the amount of ice and its capability to backscatter the visible light. Moreover, it allows to test existing microphysical

models, particularly the BV2015 model and its original representation of bulk optical properties as a function of the in-cloud temperature and IWC. The results of this study point out the overall good coherence of the BV2015 model but also its limitations in representing all the different measured profiles, especially due to the poor representation of the exact backscattering characteristics of the bulk ice.

## 5  Conclusions

In this paper a method to retrieve IWC profiles of cirrus clouds from the synergy of ground-based lidar and TIR radiometer measurements has been presented. The algorithm is based on optimal estimation theory and combines the visible lidar and TIR radiometer measurements in a common retrieval framework to retrieve profiles of IWC together with a correction factor for the backscatter intensity of bulk ice cloud particles.

As an initial step, an algorithm to retrieve IWC and extinction profiles (outside the cloud) from the lidar measurements alone was developed. Due to the backscatter-to-extinction ambiguity arising from the combination of scattering and absorption processes in the atmosphere, assumptions are required for the backscatter-to-extinction ratio and the retrieval results strongly depend on these assumptions. As a consequence, the challenge is to find ways to reduce the uncertainties in the retrieval arising from an insufficient knowledge of the backscatter-to-extinction ratio.

To overcome the backscatter-to-extinction ambiguity, we showed in a second step that it is possible to use TIR radiances to constrain the backscatter-to-extinction ratio defined as the product of the single scattering albedo and the phase function in the backscattering direction. The latter has not been yet fully characterized and is associated with large uncertainties. Moreover, it strongly depends on the characteristics of the particles composing the cloud. However, the BV2015 microphysical model links the optical properties of cirrus clouds directly to the IWC without the need for assumptions about the particle shape and PSD. It allows to obtain the single scattering albedo and the asymmetry parameter (from which the phase function is parametrized) as a function of IWC and in-cloud temperature alone. Our algorithm benefits from the fact that TIR radiances are sensitive to the integrated IWC over the whole cloud (IWP) and that the IWC of each layer governs the optical properties via the microphysical model. That means the backscatter intensity of the ice crystals is constrained by the TIR radiances under the assumption that the single scattering albedo is represented sufficiently accurately in the microphysical model. Consequently, our synergy algorithm retrieves a profile of IWC together with a correction factor for the phase function of the ice crystals in the exact backscattering direction which is assumed to be constant over the entire cloud profile. Hence, the integration of the TIR radiances into the optimal estimation framework allows to retrieve the lidar ratio although we are using backscattering profiles from a simple micro-pulse lidar.

It is important to note that the same microphysical model has been used to compute the bulk ice optical properties (i.e. the scattering and absorption coefficients as well as the asymmetry parameter and the phase function) for all wavelengths considered in this study. The consistency of this microphysical model over a large portion of the electromagnetic spectrum ranging from the visible to the infrared has been tested in numerous studies. Nevertheless, the parametrization of these optical properties as a function of IWC and temperature may introduce some uncertainty. However, a personal communication from

A. J. Baran suggests that the error introduced by such a parametrization is rather small (smaller than 5%). Thus, we believe that the results presented in this paper are robust and mainly point out the misrepresentation of the phase function in the exact backscattering direction, which is a key result of this study.

Another achievement of our algorithm is the integration of information from the whole atmospheric profile, accessible thanks to the active lidar measurements, in the forward modelling of the TIR radiances. Most common retrieval algorithms for passive sensors assume a homogeneous cloud and include only information about the cloud altitude from active measurements in the radiative transfer calculations. The synergy between the lidar and the TIR radiometer measurements established in this paper allows to account for the profile of IWC in the radiative transfer model. Furthermore, the extinction of aerosols which may be present in the atmosphere is included in the TIR forward model although further information on the aerosol type is required. In this study, the aerosol optical properties were fixed to a predefined aerosol model and an improvement of our method would be to better characterize the properties of the aerosols which are actually present during the measurement. It is worth noting that the high vertical resolution of the radiative transfer calculations in the TIR is possible thanks to the numerical efficiency of the radiative transfer model LIDORT discussed in Sect. 4.1 which allows to obtain the radiances and Jacobians for different atmospheric parameters from a single simulation.

The results for the case study discussed in Sect. 4.2 show for certain periods a quite good agreement of the retrieved lidar ratios from our synergy algorithm with the literature. When the cloud is optically very thin, the signal in the TIR radiometer measurements is very small resulting in a large uncertainty in the retrieval which seems to be rather logical. However, it is important to keep in mind that the results depend on several factors. All retrievals shown in this study were performed for a multiple scattering factor of $\eta = 0.75$ for ice clouds, and the backscatter-to-extinction ratio for aerosols was fixed to the value for an urban water soluble aerosol from the OPAC database. Changing those values may change the retrievals and further sensitivity studies with our algorithm are necessary to evaluate the effects of (1) a varying multiple scattering factor and (2) using other aerosol models.

Furthermore, when regarding ground-based TIR radiometer measurements, a good characterization of the surrounding atmosphere, especially the water vapor profile, is crucial since the TIR radiances are very sensitive to water vapor which is spatially and temporally highly variable. Hence, the ECMWF reanalysis profiles used in this study, which are available for four time steps at 0, 6, 12 and 18 UTC and have a spatial resolution of $1°$, may not be accurate enough to characterize the local water vapor profile at our measurement site during the measurement. It is certain that a better characterization of the water vapor profile, e.g. from micro-wave radiometer measurements, would help to reduce the uncertainties in our retrievals.

Finally, the quality of the measured TIR radiances plays an important role. For the case study presented here, the temperature correction of the sensitivity of the instrument results in a quite large uncertainty because of a large temperature difference between the temperatures during the measurement and the calibration. It was shown in Sect. 3.3.2 that it was not possible to simulate the clear sky radiances measured with channel C09. Hence, this channel was not included in the analysis. This might be due to a bad characterization of the spectral response function of the instrument as mentioned above and/or to an insufficient temperature correction. Thus, another improvement of our method would be to isolate the instrument from atmospheric temperature influences.

Nevertheless, the first results obtained from this algorithm are promising and we showed that our method allows to converge at the same time towards the measurements of two very different instruments. However, these results have to be confirmed in future studies for other measurement periods and measurement sites.

*Competing interests.* The authors declare that they have no conflict of interest.

5 *Acknowledgements.* The authors thank the Région Hauts-de-France, and the Ministère de l'Enseignement Supérieur et de la Recherche (CPER Climibio), and the European Fund for Regional Economic Development for their financial support. The authors thank the CaPPA project (Chemical and Physical Properties of the Atmosphere) funded by the French National Research Agency (ANR) through the PIA (Programme d'Investissement d'Avenir) under contract "ANR-11-LABX-0005-01" and by the Regional Council "Hauts-de-France" and the "European Funds for Regional Economic Development" (FEDER). The ACTRIS-FR research infrastructure is acknowledged for finacial 10 support.

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
