# Peer review of "An algorithm to retrieve ice water content profiles in cirrus clouds from the synergy of ground-based lidar and thermal infrared radiometer measurements"

_Atmospheric Measurement Techniques, 2018_

## Referee Comment (RC2)

**An algorithm to retrieve ice water content profiles in cirrus clouds from the synergy of ground-based lidar and thermal infrared radiometer measurements**

Friederike Hemmer1, Laurent C.-Labonnote1, Frédéric Parol1, Gérard Brogniez1, Bahaiddin Damiri2, and Thierry Podvin1

1Laboratoire d'Optique Atmosphérique, Université Lille, Villeneuve d'Ascq, France 2Cimel Electronique, Paris, France

Correspondence: Friederike Hemmer (friederike.hemmer@univ-lille.fr)

**Abstract.** The algorithm presented in this paper was developed to retrieve ice water content (IWC) profiles in cirrus clouds. It is based on optimal estimation theory and combines ground-based visible lidar and thermal infrared (TIR) radiometer measurements in a common retrieval framework to retrieve profiles of IWC together with a correction factor for the backscatter intensity of cirrus cloud particles. In a first step, we introduce a method to retrieve extinction and IWC profiles in cirrus clouds from the

- 5 lidar measurements alone and demonstrate the shortcomings of this approach due to the backscatter-to-extinction ambiguity. In a second step, we show that TIR radiances constrain the backscattering of the ice crystals at the visible lidar wavelength by constraining the ice water path (IWP) and hence the IWC which is linked to the optical properties of the ice crystals via a realistic bulk ice microphysical model. The scattering phase function obtained from the microphysical model has a flat ending without backscattering peak. We show that retrievals with the lidar only algorithm using this phase function in backscattering
- direction to define the backscatter-to-extinction ratio of the ice crystals result in an overestimation of IWC which is inconsistent with the TIR radiometer measurements. Hence, a synergy algorithm was developed that combines the profiles of backscattering measured by the lidar and the measurements of TIR radiances in a common optimal estimation framework to retrieve together with the IWC profile a correction factor for the phase function of the bulk ice crystals in backscattering direction. We show that this approach allows to simultaneously converge towards the measurements of two independent instruments and that the

15 first results of the retrieved lidar ratios for cirrus clouds agree with previous studies.

**1 Introduction**

The importance of clouds for the climate system has been extensively discussed during the last decades (Stephens, 2005). Although their essential role in the Earth's radiative budget is unquestionable, they still remain a major source of uncertainty in climate change estimates (Boucher et al., 2013). In particular, the important but complex impact of cirrus clouds has long

20 been recognized (Liou, 1986) but is still not well quantified nowadays. This is due to the large range of varying shapes and sizes of ice crystals composing the eloud which may interact in different ways with atmospheric radiation by scattering and absorption processes. The net radiative effect of cirrus clouds is generally positive but can be negative as well (Zhang et al.,

1999). It is determined on the one hand by its macrophysical properties, e.g. altitude, geometrical thickness, temperature and the difference between the temperature of the cloud and the surface (e.g., Stephens and Webster, 1981). On the other hand it depends on the optical properties of the cloud which are in turn governed by its microphysics, especially the size, shape and number density of particles.

- 5 Recent advances in satellite observational systems, particularly the Cloud Profiling Radar (CPR; Stephens et al. (2002)) aboard CloudSat and the Cloud-Aerosol Lidar with Orthogonal Polarization (CALIOP; Winker et al. (2009, 2010)) aboard CALIPSO as part of the international satellite constellation known as A-Train, have shown that the occurrence of cirrus clouds in the atmosphere is much higher than previously presumed (Stubenrauch et al., 2013). Mace et al. (2009) quantified the global occurrence frequencies to 40-60% whereas earlier estimates expected about 20-30% with a higher coverage of 60-70% in the
- 10 tropics (e.g., Liou, 1986; Wylie et al., 1994). This underlines the importance of studying the characteristics of cirrus clouds to estimate their influence on the radiation budget.

Lidar measurements have proven to be a powerful tool to study even the most tenuous cloud layers (e.g., Sassen, 1991). The measured backscatter profiles provide information about the cloud base and top altitudes which can be related to temperature by using atmospheric temperature profiles from model reanalysis or radiosounding. Furthermore, these measurements yield

- 15 the possibility to retrieve profiles of particle extinction and hence the optical depth of the cirrus cloud by making assumptions about the so-called backscatter-to-extinction ratio. However, there are more advanced lidar systems which do not require such assumptions that introduce large errors on the estimated cloud optical depth. Raman lidars for example can provide particle extinction directly since the inelastic Raman backscatter signal is only sensitive to extinction but not to backscattering (Ansmann et al., 1990, 1992). High-spectral resolution lidars are also capable of measuring particle extinction directly with
- 20 the help of two channels: one that measures the backscatter originating from the entire atmosphere (molecular plus particle) and one that measures only the molecular contribution by removing with an absorption filter the central portion of the signal which is associated with aerosol or cloud particles. From these two simultaneous measurements the particle extinction can be derived from the change in the slope of the molecular signal relative to a clear sky atmosphere (Turner and Eloranta, 2008). Other lidars, e.g. CALIOP, include polarization measurements from which the cloud phase can be determined since ice
- crystals tend to depolarize the incident visible radiation whereas for water droplets no such depolarization is observed (Sassen, 1991). However, most ground-based lidars operated at present are simpler systems (Campbell et al., 2015). Thus, the algorithm presented here was developed for the exploitation of data from a simple micro-pulse lidar although it might be applied to other lidars in future studies. Nowadays, it is common to operate lidar systems from space as well as from the ground and our method should be applicable to all types of lidar. 
[revised manuscript text omitted]

Field et al. (2005, 2007). This parametrization is independent of assumptions about ice crystal shape and is based on a large number of in situ measured PSDs. It does not include measurements of ice crystal size less than  $100 \,\mu\text{m}$  due to the shattering problem (Strapp et al., 2001; Field et al., 2003). For particles smaller than  $100 \,\mu\text{m}$  an exponential PSD is assumed. Field et al. (2005, 2007) found that ice crystal PSDs can be described by a single underlying PSD from which the initial PSD can be

- 5 generated with the help of two moments. They linked the second moment, which is the IWC, to any other moment via polynomial fits to the in-cloud temperature. As a consequence, the PSD can be obtained from the IWC and the in-cloud temperature alone. Baran et al. (2011, 2014a, b) constructed a large database that combines the ensemble model and the Field et al. (2005, 2007) parametrization. Based on a total number of 20662 in situ measurements of IWC and in-cloud temperature from different aircraft-based field campaigns located in the tropics and in the mid-latitudes, the PSDs were generated and for each PSD the
- 10 single scattering properties were calculated for 145 wavelengths between 0.2 and 120 µm. This ice optical property database was tested by Vidot et al. (2015) who developed a new ice cloud parametrization that predicts the bulk extinction and scattering coefficients as well as the asymmetry parameter as a function of in-cloud temperature and IWC without the need of a priori information on the shape and the effective diameter of the ice crystals. The single scattering albedo is then defined as the ratio of the scattering to the extinction coefficient, and the scattering phase function is generated from the asymmetry parameter using
- 15 the analytic phase function of Baran et al. (2001) which is a linear piecewise parametrization of the Henyey-Greenstein phase function depending only on the asymmetry parameter. It is kept smooth and featureless since atmospheric ice crystals may be distorted, roughened or contain inclusions of air or aerosols. All these processes would remove or reduce the optical features of the phase function like the halos at 22° or 46° and the backscattering peak (e.g., Macke et al., 1996; Labonnote et al., 2001). It has been demonstrated by the authors that the Baran et al. (2001) parametrization reproduces short-wave multi-angle satellite
- 20 and aicraft observations, and Baran and Francis (2004) showed a good agreement with high-resolution infrared observations between 3 and 18 µm. It has also been tested and shown to be in good agreement against the backscattering features observed from POLarization and Directionality of the Earth's Reflectances (POLDER) measurements (Baran and Labonnote, 2007). The scattering phase function, especially in the exact backscattering direction, is a crucial parameter in our algorithm since it defines the lidar backscatter-to-extinction ratio. It will be discussed in more detail in Sect. 3.3.1.
- For cirrus cloud layers, our algorithm seeks to retrieve the IWC. Thus, before the lidar equation (Eq. 1) can be applied, the IWC has to be transformed into an extinction. For this purpose, the microphysical model described above is used. The ice cloud parametrization of Vidot et al. (2015) provides the link between the desired parameter (IWC) and the quantities required for the inversion of the lidar equation which are the extinction coefficient, the single scattering albedo and the asymmetry parameter from which the phase function is generated using the parametrization of Baran et al. (2001). For the remainder of this paper, we
- 30 will call this microphysical model BV2015. The necessary temperature information is obtained from the European Centre for Medium-Range Weather Forecasts (ECMWF) reanalysis by matching atmospheric temperature profiles to the corresponding cirrus cloud altitude.

**3.2 Inversion method**

10

As mentioned above, we apply an optimal estimation method to invert the lidar equation following Stephens et al. (2001). Optimal estimation is based on a bayesian approach which uses probability density functions to link the measurement space to the state space accounting for their uncertainties (Rodgers, 2000). This approach allows to find the most likely solution which

5 is consistent with both the measurement and any given prior knowledge of the state within the range of their uncertainties. In general, the measurement vector y can be related to the state vector x via the forward model **F** by

$$\boldsymbol{y} = \mathbf{F}(\boldsymbol{x}) + \boldsymbol{\epsilon},\tag{4}$$

[revised manuscript text omitted]

$$\mathbf{X} = \begin{pmatrix} \frac{\partial F_{1}}{\partial \sigma_{p,1}} & \cdots & \frac{\partial F_{1}}{\partial \sigma_{p,j,bn-1}} & \frac{\partial F_{1}}{\partial W C_{j,bn}} & \cdots & \frac{\partial F_{1}}{\partial W C_{j,lop}} & \frac{\partial F_{1}}{\partial \sigma_{p,j,lop+1}} & \cdots & \frac{\partial F_{1}}{\partial \sigma_{p,N}} \\ \frac{\partial F_{2}}{\partial \sigma_{p,1}} & \cdots & \frac{\partial F_{2}}{\partial \sigma_{p,j,bn-1}} & \frac{\partial F_{2}}{\partial W C_{j,bn}} & \cdots & \frac{\partial F_{2}}{\partial W C_{j,lop}} & \frac{\partial F_{2}}{\partial \sigma_{p,j,lop+1}} & \cdots & \frac{\partial F_{2}}{\partial \sigma_{p,N}} \\ \vdots & \cdots & \vdots & \vdots & \cdots & \vdots & \vdots & \cdots & \vdots \\ \frac{\partial F_{j,bn-1}}{\partial \sigma_{p,1}} & \cdots & \frac{\partial F_{j,bn-1}}{\partial \sigma_{p,j,bn-1}} & \frac{\partial F_{j,bn-1}}{\partial W C_{j,bn}} & \cdots & \frac{\partial F_{j,bn-1}}{\partial W C_{j,lop}} & \frac{\partial F_{j,bn-1}}{\partial \sigma_{p,N}} & \cdots & \frac{\partial F_{j,bn-1}}{\partial \sigma_{p,N}} \\ \vdots & \cdots & \vdots & \vdots & \cdots & \vdots & \vdots & \cdots & \vdots \\ \frac{\partial F_{j,bn}}{\partial \sigma_{p,1}} & \cdots & \frac{\partial F_{j,bn}}{\partial \sigma_{p,j,bn-1}} & \frac{\partial F_{j,bn}}{\partial W C_{j,bn}} & \cdots & \frac{\partial F_{j,bn}}{\partial W C_{j,lop}} & \frac{\partial F_{j,bn}}{\partial \sigma_{p,N,lop+1}} & \cdots & \frac{\partial F_{j,bn}}{\partial \sigma_{p,N}} \\ \vdots & \cdots & \vdots & \vdots & \cdots & \vdots & \vdots & \cdots & \vdots \\ \frac{\partial F_{j,lop}}{\partial \sigma_{p,1}} & \cdots & \frac{\partial F_{j,lop}}{\partial \sigma_{p,j,bn-1}} & \frac{\partial F_{j,lop}}{\partial W C_{j,bn}} & \cdots & \frac{\partial F_{j,lop}}{\partial W C_{j,lop}} & \frac{\partial F_{j,lop}}{\partial \sigma_{p,j,lop+1}} & \cdots & \frac{\partial F_{j,lop}}{\partial \sigma_{p,N}} \\ \vdots & \cdots & \vdots & \vdots & \cdots & \vdots & \vdots & \cdots & \vdots \\ \frac{\partial F_{j,lop+1}}{\partial \sigma_{p,1}} & \cdots & \frac{\partial F_{j,lop+1}}{\partial W C_{j,bn}} & \frac{\partial F_{j,lop+1}}{\partial W C_{j,lop}} & \frac{\partial F_{j,lop+1}}{\partial \sigma_{p,j,lop+1}} & \cdots & \frac{\partial F_{j,lop}}{\partial \sigma_{p,N}} \\ \vdots & \cdots & \vdots & \vdots & \cdots & \vdots & \vdots & \cdots & \vdots \\ \frac{\partial F_{N}}{\partial \sigma_{p,1}} & \cdots & \frac{\partial F_{N}}{\partial \sigma_{p,j,bn-1}} & \frac{\partial F_{N}}{\partial W C_{j,bn}} & \cdots & \frac{\partial F_{N}}{\partial W C_{j,lop}} & \frac{\partial F_{N}}{\partial \sigma_{p,N}} \end{pmatrix} \end{pmatrix}$$

where the terms  $F(x_j, b_j)$ ,  $\sigma_p(r_j)$  and IWC $(r_j)$  have been shortened to  $F_j$ ,  $\sigma_{p,j}$  and IWCj, respectively, to increase the readability. This short notation will be used for all variables which are a function of range for the remainder of this article. Inside cirrus cloud layers, the partial derivatives are expressed by

$$\mathbf{K}_{ij} = \frac{\partial \mathbf{F}_i}{\partial \mathbf{IW} \mathbf{C}_j} = \frac{\partial \mathbf{F}_i}{\partial \sigma_{p,j}} \frac{\partial \sigma_{p,j}}{\partial \mathbf{IW} \mathbf{C}_j},\tag{13}$$

[revised manuscript text omitted]

---

## Referee Comment (RC1) · Anonymous Referee #1 · 7 Nov 2018

**Manuscript number: amt-2018-320**

Full title: An algorithm to retrieve ice water content profiles in cirrus clouds from the synergy of ground-based lidar and thermal infrared radiometer measurements

Author(s): Hemmer et al.

This paper demonstrates an algorithm for ice water content profile retrievals from ground-based lidar and thermal infrared radiometer measurements on an optimal estimation framework. The bulk optical properties rely on a parameterization that links the optical properties with the ice water content and temperature. At first, the shortcoming of the retrieval method based on lidar alone measurements focusing on the uncertainty in the lidar ratio is demonstrated. Then, the authors show that the combined retrieval method based on lidar and thermal infrared measurements benefits the ice water content profile retrievals by reducing the uncertainty due to the lidar ratio. The computed optical thickness from retrieved ice water content profile generally similar for optimally moderately thick clouds and is underestimated compared to the counterparts based on the two-way transmissivity approach.

Overall, this paper is well written and well organized. The methods are sound. The topic presented in this paper is suitable to Atmospheric Measurement Techniques. I recommend the paper for publication, once the several points below have been taken care of.

**General comments**

The current manuscript contains several grammatical errors and redundant descriptions. I recommend the authors to proofread the manuscript again and encourage to make the redundant descriptions shorter. These treatments may help readers understand the contents.

**Minor comments**

1. Pages 4–5: The authors should specify the following the instrumental characteristics: (1) The lidar pointing zenith angle; and (2) the FOV of the thermal infrared radiometer. If the FOV is not small enough, does the inhomogeneity of the

response function along with viewing zenith angles affects the uncertainty in thermal infrared radiometric signals?

- 2. Pages 7–8 "These two parameters are obtained for each cloud...": These two paragraphs are confusing. The first paragraph mentions that the single scattering albedo and the phase function are obtained from the ice model introduced by Baran and Labonnote (2007). However, the second paragraph mentions that Vidot et al. (2015) parameterization is used to link the ice water content with several parameters including the extinction coefficient, single scattering albedo, and asymmetry parameter based on Baran et al. (2001), which is inconsistent with Baran and Labonnote (2007). I'm not sure if I understand these two paragraphs correctly. The inconsistent optical properties may arise an uncertainty, and the authors should clarify this. In addition, please make the first paragraph shorter. In the paragraph, although the authors introduce many parameterizations regarding ice properties, the paper only use the BV2015 parameterization.
- 3. Page 9, Equation 8: Should "<<" be "\le " or "<"?
- 4. Page 10: Please add the descriptions about the Jacobian K if it is the case that opaque cirrus clouds (lidar signals cannot reach to cloud top) are present. You may not have  $\partial F_{j_{top}}/\partial IWC_{j_{top}}$ .
- 5. Page 19, Line 4 "*the COT decreases and with it the simulated radiances*": Does this include a typo? Could you rephrase it?
- 6. Page 27 Figure 12 (e): Could you please reconsider the colors for the measurements? It is hard to recognize these plots.
- 7. Page 28, Lines 7–19: It is unfair to compare different qualities (i.e., COT and an effective COT). Since the author assumes the multiple scattering factor to be 0.75 for ice clouds throughout the paper, you can compare COT from the combined method with COT converted from an effective COT. In Figure 12c, although the effective COT is smaller than actual COT by 33% (if the multiple scattering factor =

0.75), I notice that the uncertainty due to multiple scattering factor cannot fully explain the underestimated COT from the combined method, particularly during UTC 16–17. The estimated lidar ratios are in the reasonable range (i.e., 20–40 sr) during the period. However, the effective COT from the two-way method is larger than the COT from the combined method by a factor of 2–3, and the multiple scattering factor of 0.3–0.5, which would compensate for the large effective COT, is unrealistic for ground-based lidar measurements. Therefore, I suggest the authors to add discussions in the paragraph regarding other potential sources that cause underestimated COT from the combined method. It may be good to mention a potential bias in the thermal infrared radiometer due to temperature.

---

## Author Comment (AC1) · 1 Feb 2019

**Reply to Anonymous Referee #1**

The authors like to thank anonymous referee #1 for his critical and constructive remarks that helped to improve our manuscript.

Manuscript number: amt-2018-320

Full title: An algorithm to retrieve ice water content profiles in cirrus clouds from the synergy of ground-based lidar and thermal infrared radiometer measurements

Author(s): Hemmer et al.

This paper demonstrates an algorithm for ice water content profile retrievals from ground-based lidar and thermal infrared radiometer measurements on an optimal estimation framework. The bulk optical properties rely on a parameterization that links the optical properties with the ice water content and temperature. At first, the shortcoming of the retrieval method based on lidar alone measurements focusing on the uncertainty in the lidar ratio is demonstrated. Then, the authors show that the combined retrieval method based on lidar and thermal infrared measurements benefits the ice water content profile retrievals by reducing the uncertainty due to the lidar ratio. The computed optical thickness from retrieved ice water content profile generally similar for optimally moderately thick clouds and is underestimated compared to the counterparts based on the two-way transmissivity approach.

Overall, this paper is well written and well organized. The methods are sound. The topic presented in this paper is suitable to Atmospheric Measurement Techniques. I recommend the paper for publication, once the several points below have been taken care of.

**General comments**

The current manuscript contains several grammatical errors and redundant descriptions. I recommend the authors to proofread the manuscript again and encourage to make the redundant descriptions shorter. These treatments may help readers understand the contents.

Efforts have been made to correct grammatical errors and shorten redundant descriptions.

**Minor comments**

1. Pages 4–5: The authors should specify the following the instrumental characteristics: (1) The lidar pointing zenith angle; and (2) the FOV of the thermal infrared radiometer. If the FOV is not small enough, does the inhomogeneity of the response function along with viewing zenith angles affects the uncertainty in thermal infrared radiometric signals?

In this study, the lidar was pointing directly vertical with a zenith angle of 0°. The FOV of the thermal infrared radiometer is 3.5°. This information has been added to the text.

Hence, the FOV of the radiometer is larger than the FOV of the lidar (55 μrad). Therefore, the two instruments do not see exactly the same cloud area. This difference also depends on the altitude of the cloud. As in almost all remote sensing algorithms, we have assumed a homogeneous cloud in the instrument FOV and did not take into account any uncertainty due to sub-pixel heterogeneity. This assumption has been added to the text.

Furthermore, the information about the ground-based version of the TIR radiometer has been removed since it is irrelevant for this study.

2. Pages 7–8 "These two parameters are obtained for each cloud…": These two paragraphs are confusing. The first paragraph mentions that the single scattering albedo and the phase function are obtained from the ice model introduced by Baran and Labonnote (2007). However, the second paragraph mentions that Vidot et al. (2015) parameterization is used to link the ice water content with several parameters including the extinction coefficient, single scattering albedo, and asymmetry parameter based on Baran et al. (2001), which is inconsistent with Baran and Labonnote (2007). I'm not sure if I understand these two paragraphs correctly. The inconsistent optical properties may arise an uncertainty, and the authors should clarify this. In addition, please make the first paragraph shorter. In the paragraph, although the authors introduce many parameterizations regarding ice properties, the paper only use the BV2015 parameterization.

Thank you for this remark. We tried to explain the microphysical in detail which might have resulted in a confusing description. Indeed, the optical properties are obtained from the Vidot et al (2015) parametrization. However, this parametrization is based on the microphysical model of Baran and Labonnote (2007) so there is no inconsistency here. Efforts have been made to shorten this paragraph and give the important information more directly to help the reader to understand the underlying microphysical assumptions of our methodology.

3. Page 9, Equation 8: Should "<<" be "≤" or "<"?

The symbol "<<" was chosen to indicate that we stop the iteration when the left-hand side of Eq. 8 is much smaller than the size of the measurement vector. However, it has been changed to "<" in the manuscript.

4. Page 10: Please add the descriptions about the Jacobian K if it is the case that opaque cirrus clouds (lidar signals cannot reach to cloud top) are present. You may not have $\partial F_{j\_top}/\partial IWC_{j\_top}$.

If the lidar signal is completely attenuated, the size of the measurement vector and consequently the size of the state vector are reduced. In this case, only the altitudes until full attenuation of the lidar signal are considered. Thus, the Jacobian is also reduced and contains

N_att lines (and columns) where N_att is the number of levels until the level of full attenuation. This information has been included in Sect. 3.2.
However, for the case study presented in this paper this case is less relevant since the lidar signal reached the cloud top for almost all treated profiles. Only in the end of the considered period around 19.8 UTC the signal was completely attenuated. For the according profiles, the lidar only algorithm as well as the synergy algorithm do not converge. This could be related to the reduced Jacobian but we believe that this non-convergence is more likely due to physical reasons since the cloud base at this time was located in low altitudes (around 6 km) and the temperature was rather warm for a cirrus cloud (between 245 and 250 K). In this temperature range, the presence of supercooled liquid droplets is possible which is not included in the BV2015 microphysical model. Hence, this model does probably not represent the optical properties of this cloud accurately enough. In order to draw more sophisticated conclusions, more cases have to be analyzed where the lidar profile was completely attenuated.

5. Page 19, Line 4 "the COT decreases and with it the simulated radiances": Does this include a typo? Could you rephrase it?

What we wanted to say is that the COT decreases which causes the simulated radiances to decrease as well. The sentence has been rephrased.

6. Page 27 Figure 12 (e): Could you please reconsider the colors for the measurements? It is hard to recognize these plots.

The colors for the measurements in plot 12e have been changed in order to increase the readability of this plot.

7. Page 28, Lines 7–19: It is unfair to compare different qualities (i.e., COT and an effective COT). Since the author assumes the multiple scattering factor to be 0.75 for ice clouds throughout the paper, you can compare COT from the combined method with COT converted from an effective COT. In Figure 12c, although the effective COT is smaller than actual COT by 33% (if the multiple scattering factor = 0.75), I notice that the uncertainty due to multiple scattering factor cannot fully explain the underestimated COT from the combined method, particularly during UTC 16–17. The estimated lidar ratios are in the reasonable range (i.e., 20–40 sr) during the period. However, the effective COT from the two-way method is larger than the COT from the combined method by a factor of 2–3, and the multiple scattering factor of 0.3–0.5, which would compensate for the large effective COT, is unrealistic for ground-based lidar measurements. Therefore, I suggest the authors to add discussions in the paragraph regarding other potential sources that cause underestimated COT from the combined method. It may be good to mention a potential bias in the thermal infrared radiometer due to temperature.

Thank you for this remark. In fact, the COT from the transmission method plotted in Fig. 12c has already been corrected by the multiple scattering factor as mentioned in Sect. 4.2 (p. 28,

lines 10-12 (page and line numbers refer to the discussion paper): "… the COT from the transmission method reported in Fig. 12c has been divided by the assumed multiple scattering factor for ice clouds of $\eta = 0.75$ in order to be consistent with the retrievals from the synergy algorithm which have been performed for $\eta = 0.75$ as well."). This phrase has been reformulated in order to help the reader to better understand. However, your remark that the assumed multiple scattering factor alone cannot explain the discrepancy between our synergy algorithm and the transmissivity approach is totally justified. Hence, we included a further discussion of this point:

"However, the multiple scattering factor alone cannot explain the inconsistency between the COT retrieved with the synergy algorithm and the COT derived from the transmission method. Another possible reason for this discrepancy may arise from the uncertainty in the transmission method itself because it depends on a good characterization of the molecular signal above the cloud and a good estimation of the cloud top altitude. These parameters are related with rather large uncertainties due to the quite noisy micro-pulse lidar signal in the high altitudes of cirrus clouds. Furthermore, the discrepancy between the two COTs could also originate from a potential bias in the TIR radiometer measurements due to an inaccurate temperature correction as mentioned in Sect. 4.1, or from a potential bias in the TIR forward model due to an inaccurate description of the atmospheric water vapor profile since the TIR radiometer measurements are very sensitive to water vapor (Dubuisson et al., 2008). Finally, the difference in the COTs from the synergy algorithm and the transmission method could also originate from the microphysical model which might not be perfect. The extinction at the lidar wavelength, which is calculated based on the IWP constrained by the TIR radiometer measurements, could be slightly underestimated. Figure 12c shows that if the IWP is larger, the difference between the COTs from the two methods becomes smaller. This can be explained by the fact that the contribution of the water vapor in the TIR radiometer measurements is more important for thin clouds than for thick clouds leading to an underestimation of the IWP and consequently an underestimation of the extinction, especially in case of thin cirrus clouds."

---

## Author Comment (AC2) · 1 Feb 2019

**Reply to Anonymous Referee #2**

The authors like to thank anonymous referee #2 for his detailed remarks and corrections of the manuscript that helped to improve the readability of our paper.

**General Comments**

To be brief, I agree with Reviewer 1's assessment. In particular, with their last point 7. To this point, I would further note that I could not find the fov of the lidar used nor the laser divergence. Such parameters should be listed as they directly impact the magnitude of the multiple-scattering effects in the lidar signals.

To play devil's advocate, it could be supposed that the important results presented in this paper all depend on the accuracy of the microphysical model used in the optimal estimation retrieval. Specifically, if the extinction at the lidar wavelength for a given IWC predicted by the model is inaccurate then the inversion results will be inaccurate, leading to wrong values of the retrieved lidar-ratio. However, the authors should point out that this is not quite the case. What is important is the combined lidar + radiometer retrieval is the relationship between the TIR absorption optical depth profile and the lidar extinction profile implied by the microphysical model. One of Platt's earlier papers discussed this important relationship. This relationship is expected to be robust for a large range of particle shapes and sizes. So I believe the results presented in this paper are likely at least robust with respect to inaccuracies in the microphysical model (at least with respect to the relationship between the extinction, IWC and temperature). This point should be covered in the discussion section of the paper and likely mentioned in the conclusions.

Thank you for this comment. We agree that the discussion of the microphysical model could have been more detailed in our discussion paper. Hence, we added the following paragraph to the conclusion section (see also the paragraph below that has been added to Sect 3.2 dealing with the error characterization of the microphysical model which is related to this point):

"It is important to note that the same microphysical model has been used to compute the bulk ice optical properties (i.e. the scattering and absorption coefficients as well as the asymmetry parameter and the phase function) for all wavelengths considered in this study. The consistency of this microphysical model over a large portion of the electromagnetic spectrum ranging from the visible to the infrared has been tested in numerous studies. Nevertheless, the parametrization of these optical properties as a function of IWC and temperature may introduce some uncertainty. However, a personal communication from A. J. Baran suggests that the error introduced by such a parametrization is rather small (smaller than 5%). Thus, we believe that the results presented in this paper are robust and mainly point out the misrepresentation of the phase function in the exact backscattering direction, which is a key result of this study."

**Specific Comments (page and line numbers refer to the discussion paper)**

p. 2, lines 28-29:
*"Nowadays, it is common to operate lidar systems from space as well as from the ground and our method should be applicable to all types of lidar."*
Somewhat vague sentence. Please re-write. It would be better to be more direct. e.g. The algorithm presented here should be applicable to combined TIR and simple backscatter lidar measurements, included space-based observations.

Thank you for this suggestion, the sentence has been re-written:
"Our method should be applicable to combined TIR and simple backscatter lidar measurements from ground-based as well as space-based observations."

p. 4, line 2:
"dispose of" means "get rid of". I do not think you want to say this here.

What we wanted to say is "possess", it has been corrected.

p. 4, lines 27-32:
Since you later consider multiple-scattering effect, it would be useful to state the laser divergence and the fov of the lidar receiver somewhere in this section.

You are right, this information has been missing and was added to the paragraph:
"The divergence of the laser beam as well as the field of view (FOV) of the receiver are both 55 µrad."

p. 9, line 7:

What do you do if noise causes the power to be zero or negative?

First of all, we apply a binomial filter to the measured lidar profiles to smooth the signal. If the power still becomes zero or negative, the profile is treated until the altitude just below the first occurrence of a negative power. In general, this only occurs above the cloud top but if this is happening in lower levels, the data is considered to be of insufficient quality and the profiles are excluded (this was never the case for the considered time period in this study). In any case, we stop the retrieval at the latest 500 m above the retrieved cloud top as mentioned on p. 14, lines 7-8.

p. 11, line 20:

Here you are calculating Se assuming that the extinction-vs-IWC relationship from the microphysical model is perfect. i.e. you are NOT taking into account this component of the model error. This is OK. However, you should make it clear in the discussion that you are neglecting this component and why you are doing this (too complex ?, errors in microphysical model not known etc.. ?)

Indeed, we are currently not taking into account an error related to the microphysical ice cloud model. This error is important but it is also difficult to quantify. We plan to evaluate it for future versions of our algorithm. For this paper, the following paragraph has been added to the discussion of the variance-covariance matrix:

"It should be noted that the characterization of the errors related to the BV2015 parametrization is very challenging. Thus, no error for the microphysical model is currently taken into account. However, this issue needs to be addressed in future studies and an evaluation of the uncertainty arising from the Vidot et al. (2015) parametrization, which has to be integrated in future versions of our algorithm, is planned. This evaluation could be performed by comparing the single scattering properties calculated directly from the ensemble model of Baran and Labonnote (2007) with the results from the Vidot et al. (2015) parametrization. Unfortunately, this is very costly to realize for all couples of IWC and temperature, particularly since the parametrization has not been developed by us. Furthermore, the uncertainty arising from this parametrization is assumed to be smaller than 5% (A. J. Baran, personal communication). Consequently, the present study neglects an error related to the microphysical model."

p. 17, line 13:

Is this not the other way around. That the retrieved integrated IWC depends strongly on the backscatter-to-extinction ratio?

Yes, you are right. It is indeed the other way around and has been corrected in the text.

p. 19, line 3:
"which is related to a correction factor for the phase function in backscattering direction"
Unnecessary. You have explained this numerous times before.

Okay, this half-sentence has been removed.